# An investigation of weighting schemes suitable for incorporating large ensembles into multi-model ensembles

Anna Louise Merrifield[1], Lukas Brunner[1], Ruth Lorenz[1], Iselin Medhaug[1], and Reto Knutti[1]

[1]Institute for Atmospheric and Climate Science, ETH Zurich, Zurich, Switzerland

**Correspondence:** Anna Louise Merrifield (anna.merrifield@env.ethz.ch)

**Abstract.** Multi-model ensembles can be used to estimate uncertainty in projections of regional climate, but this uncertainty often depends on the constituents of the ensemble. The dependence of uncertainty on ensemble composition is clear when single model initial condition large ensembles (SMILEs) are included within a multi-model ensemble. SMILEs allow for the quantification of internal variability, a non-negligible component of uncertainty on regional scales, but also may serve to inappropriately narrow uncertainty by giving a single model many additional votes. In advance of the mixed multi-model, SMILE Coupled Model Intercomparison version 6 (CMIP6) ensemble, we investigate weighting approaches to incorporate 50-members of the Community Earth System Model (CESM1.2.2-LE), 50-members of the Canadian Earth System Model (CanESM2-LE), and 100-members of the MPI Grand Ensemble (MPI-GE) into an 88-member Coupled Model Intercomparison Project Phase 5 (CMIP5) ensemble. The weights assigned are based on ability to reproduce observed climate (performance) and scaled by a measure of redundancy (dependence). Surface air temperature (SAT) and sea level pressure (SLP) predictors are used to determine the weights, and relationships between present and future predictor behavior are discussed. Estimated residual thermodynamic trend is proposed as an alternative predictor to replace 50-year regional SAT trends, which are more susceptible to internal variability.

Uncertainty in estimates of Northern European winter and Mediterranean summer end-of-century warming is assessed in a CMIP5 and a combined SMILE-CMIP5 multi-model ensemble. Five different weighting strategies to account for the mix of initial condition (IC) ensemble members and individually represented models within the multi-model ensemble are considered. Allowing all multi-model ensemble members to receive either equal weight or solely a performance weight (based on root-mean-square-error (RMSE) between member and observations over nine predictors) is shown to lead to uncertainty estimates that are dominated by the presence of SMILEs. A more suitable approach includes a dependence assumption, scaling either by $1/N$, the number of constituents representing a "model", or by the same RMSE distance metric used to define model performance. SMILE contributions to the weighted ensemble are smallest ($< 10\%$) when a "model" is defined as an IC ensemble, and increase slightly ($< 20\%$) when the definition of "model" expands to include members from the same institution/development stream. SMILE contributions increase further when dependence is defined by RMSE (over nine predictors) amongst members because RMSEs between SMILE members can be as large as RMSEs between SMILE members and other models. We find that an alternative RMSE distance metric, derived from global SAT and hemispheric SLP climatology, is able to better identify IC member in general and SMILE members in particular as members of the same model. Further, more subtle dependencies associated with resolution differences and component similarities are also identified by the global predictor set.

## 1   Introduction

Projections of regional climate change are both key to climate adaptation policy and fundamentally uncertain due to the nature of the climate system (Deser et al., 2012; Kunreuther et al., 2013). In order to represent regional climate uncertainty to policy-makers, scientists often turn to multi-model ensembles to provide a range of plausible outcomes a region may experience (Tebaldi and Knutti, 2007). Uncertainty in a multi-model ensemble is commonly estimated from the ensemble spread, which can be represented e.g., as the 5-95% likely range of the distribution and is usually presented with respect to the arithmetic

ensemble mean (e.g. Collins et al., 2013). This representation of uncertainty appears unambiguous, but is perhaps deceptively so. It is influenced by choices made in multi-model ensemble construction, choices that are often overlooked (Knutti et al., 2010a, b).

Multi-model ensembles, such as those constructed from Coupled Model Intercomparison Projects or CMIPs (Meehl et al., 2000), tend to be comprised of both different models and multiple members of the same model, subject to the same radiative

forcing pathway intended to reflect plausible future emissions scenario (van Vuuren et al., 2011; O'Neill et al., 2014). This choice allows the multi-model ensemble to represent two types of regional-scale uncertainty: model uncertainty and internal variability (e.g. Hawkins and Sutton, 2009; Deser et al., 2012). Model uncertainty accounts for differences in how models simulate climate, from how the equations governing flow in the atmosphere are numerically solved to how sub-grid scale processes in the climate system are parameterized. Sub-grid scale processes are often the product of complex interactions and

feedbacks between the land surface, ocean, cryosphere, and atmosphere, many of which can not be directly measured (e.g. Seneviratne et al., 2010; Deser et al., 2015). How models estimate these interactions can result in various advantages and limitations in regional climate representation, and thus affect regional uncertainty estimates.

By considering differences in regional "performance", it becomes clear that uncertainty is affected by the assumption that each member of a multi-model ensemble is an equally plausible representation of observed climate. Known biases associated

with cloud processes, land-atmosphere interactions, and sea surface temperature (e.g. Boberg and Christensen, 2012; Li and Xie, 2012; Pithan et al., 2014; Merrifield and Xie, 2016) may result in more uncertainty in projections of future climate than is warranted given our understanding of the climate system (Vogel et al., 2018). Using expert judgement to weight or select multi-model ensemble members based on process- or region-specific metrics of performance has been shown to justifiably constrain uncertainty in other studies (e.g. Abramowitz et al., 2008; Knutti et al., 2017; Lorenz et al., 2018).

The second type of uncertainty, internal variability, reflects the regional influence of the amalgamation of unpredictable fluctuations in the climate system (Hawkins and Sutton, 2009; Deser et al., 2012; Knutti and Sedláček, 2013). Internal variability is ostensibly a feature of the climate system and manifests itself in climate variables, such as regional surface air temperature (SAT), through a complex set of controlling influences, chief among them being variability in the attendant atmospheric circulation (Wallace et al., 1995, 2015; Branstator and Teng, 2017). The influence of internal atmospheric variability on SAT

can be quantified and accounted for in projections of future climate using dynamical adjustment methods (e.g. Deser et al.,

2016; Sippel et al., 2019). Additionally, internal variability can be explicitly represented by sets of simulations from the same model, subject to identical forcing, where members differ only by initial conditions (e.g. Kay et al., 2015; Maher et al., 2019). These single model initial condition large ensembles or SMILEs have become an indispensable tool to concisely represent uncertainty within a model, information that should be considered in a multi-model ensemble context (Rondeau-Genesse and

Braun, 2019).

The prospect of including SMILE members into a multi-model ensemble directly challenges another assumption that tends to be made when calculating probabilistic estimates from multi-model ensembles: each member is an independent representation of climate. Though all members of a multi-model ensemble describe the same climate system, differences in model structure and internal variability create a distribution of regional climate change estimates. Differences in model structure are often

welcome; for many applications, distributions comprised of several models are hypothesized to reflect the range of possible climate outcomes better than distributions from a single model (Abramowitz et al., 2019). When different models (that are deemed independent from each other) agree, there is a notion of robustness and increased certainty in the outcome. Ultimately, there is also a notion that as models improve, there will be a "convergence to reality" with models independently simulating the same "right" outcome.

In reality, however, members of a multi-model ensemble are often dependent entities. Narrowing of uncertainty comes through redundant representation of historical and future climate, rather than through the independent simulation of the "right" outcome (Herger et al., 2018). Redundancy within a multi-model ensemble can arise from different models having similar biases with respect to observations. Models have historically shared code, from parametrization schemes to full components, and tend to have the same limitations associated with resolution (i.e., simplified topography) (Masson and Knutti, 2011; Knutti

et al., 2013; Boé, 2018). These commonalities can cause similar climate trajectories amongst models with different names, complicating the notion of "convergence to reality" through dependence of differently named models. Another clear contributor to redundancy is multiple initial condition (IC) ensemble members that project climate trajectories which only differ by internal variability; similar trajectories are likely to exist amongst the 50 to 100 members of a SMILE. It is therefore important when assembling a multi-model ensemble that uncertainty estimates reflect that not every member is an independent entity (Pennell

and Reichler, 2011).

What constitutes an independent entity within a multi-model ensemble remains a topic of debate (Annan and Hargreaves, 2017; Abramowitz et al., 2019). Independence can be decided *a priori*, i.e., that a model, as defined by its name, is an independent entity. This choice renders IC members dependent. It could also be decided that only models from different institutions of origin are independent entities, as in the "same-center hypothesis" explored by Leduc et al. (2016). In the absence of knowl-

edge of model origin and development, independent entities could instead be defined using statistical properties of model outputs (Masson and Knutti, 2011; Bishop and Abramowitz, 2013). In this *a posteriori* definition, models may have a degree of independence, rather than simply an independent or dependent designation (Knutti et al., 2017).

Regardless of how dependent/independent entities are defined, it is important that dependence is accounted for and redundancy mitigated in order to avoid an overconfident, inappropriately narrow distribution of future change (Leduc et al., 2016;

Abramowitz et al., 2019). Dependent information reduction can be achieved through subsetting, where information deemed

dependent is discarded, or through a weighting scheme, where information is scaled by degree of dependence. In this study, we evaluate if a performance and independence weighting scheme (Knutti et al., 2017; Lorenz et al., 2018; Brunner et al., 2019) can be used to include three SMILEs into a CMIP5 multi-model ensemble and provide a justifiably constrained estimate of European regional end-of-century warming uncertainty. Northern European winter and Mediterranean summer SAT changes between the 1990-2009 and 2080-2099 mean states are considered. We discuss details of the weighting method including emergent predictor relationships and optimal parameter choices for attempting to comprehensively characterize member performance while separating independent information from information known to have common origin (SMILE members). We highlight a new metric, estimated residual thermodynamic trend, which can be used as an alternative to trend-based metrics that do not optimally reflect a model's performance on regional scales. We compare how five different weighting strategies, based on different dependence assumptions, constrain uncertainty in a CMIP5 multi-model ensemble with and without the SMILEs included. Weighted SMILE contributions in each CMIP5-SMILE "ALL" ensemble are explicitly computed. The five weighting strategies come from the continuum of assumptions that can arise in multi-model ensemble construction: (1) all members are independent and equally plausible (equal weighting), (2) some members are more realistic than others (performance weighting), (3,4) members from the same model are dependent (1/N scaling, N being number of IC members or modelling center contributions), and (5) all members are dependent to some degree (RMSE distance metric scaling). For the last approach, we demonstrate that an RMSE independence scaling that groups SMILE members and distinguishes them from other models can be obtained using large-scale, long term SAT and sea level pressure (SLP) climatology fields. The SMILEs, CMIP5, and observational datasets used in the weightings are described in Section 2, while the weighting schemes are detailed in Section 3. The influence of SMILE inclusion on the weighting under different dependence assumptions and the predictor set that identifies SMILE members as dependent entities based on RMSE distance are discussed in Section 4. To close, conclusions and discussion is presented in Section 5.

## 2 Data

The multi-model ensemble used in this study is comprised of members from the CMIP5 archive and three SMILEs: a 50-member ensemble generated using the Community Earth System Model version 1.2.2 (CESM1.2.2-LE), the 50-member Canadian Earth System Model version 2 large ensemble (CanESM2-LE), and the 100-member Max Planck Institute for Meteorology Grand Ensemble (MPI-GE). This combined CMIP5-SMILE ensemble is summarized in Table 1, which lists the name of each model and the members used. A similar CMIP5 multi-model ensemble was used in Lorenz et al. (2018) and Brunner et al. (2019) and features 88 members from 40 (named) model setups, including 13 initial condition ensembles ranging from 2 to 10 members. Additionally, for the GISS-E2-H and GISS-E2-R experiments, NASA GISS provides members from three physics-version ("p") setups that differ in atmospheric composition (AC) and aerosol indirect effects (AIE) (Miller et al., 2014). We treat the three setups as follows: p1 (prescribed AC and AIE) and p3 (prognostic AC and partial AIE) members are treated as 2 member IC ensembles and the p2 member (prognostic AC and AIE) is treated as a single member representation (Table 1). In Table 1, IC ensembles are indicated in italic and SMILEs are indicated in bold with a star. Horizontal lines denote mod-

**Table 1.** Summary of the CMIP5 + SMILE multi-model ensemble used in this study. IC ensembles within CMIP5 are indicated in italics. SMILEs are indicated in bold with a star. Modelling center/development streams groupings are separated by horizontal lines.

| Group | Model | Members Used | Group | Model | Members Used |
|---|---|---|---|---|---|
| ACCESS | ACCESS1-0 | r1i1p1 | NASA GISS | GISS-E2-R-CC | r1i1p1 |
| | ACCESS1-3 | r1i1p1 | (cont.) | *GISS-E2-R* | *r(1-2)i1p1* |
| | BNU-ESM | r1i1p1 | | GISS-E2-R | r1i1p2 |
| NCAR | *CCSM4* | *r(1-6)i1p1* | | *GISS-E2-R* | *r(1-2)i1p3* |
| | CESM1-BGC | r1i1p1 | MOHC | HadGEM2-AO | r1i1p1 |
| | *CESM1-CAM5* | *r(1-3)i1p1* | | HadGEM2-CC | r1i1p1 |
| | **CESM1.2.2-LE\*** | **r(0-49)i1p1** | | *HadGEM2-ES* | *r(1-4)i1p1* |
| CMCC | CMCC-CESM | r1i1p1 | IPSL | *IPSL-CM5A-LR* | *r(1-3)i1p1* |
| | CMCC-CMS | r1i1p1 | | IPSL-CM5A-MR | r1i1p1 |
| | CMCC-CM | r1i1p1 | | IPSL-CM5B-LR | r1i1p1 |
| | *CNRM-CM5* | *r(1,2,4,6,10)i1p1* | MIROC | MIROC-ESM | r1i1p1 |
| | *CSIRO-Mk3-6-0* | *r(1-10)i1p1* | | MIROC-ESM-CHEM | r1i1p1 |
| CCCma | **CanESM2-LE\*** | **r(1-50)i1p1** | | *MIROC5* | *r(1-3)i1p1* |
| | *CanESM2* | *r(1-5)i1p1* | MPI-M | *MPI-ESM-LR* | *r(1-3)i1p1* |
| | *EC-EARTH* | *r(1,2,8,9,12)i1p1* | | MPI-ESM-MR | r1i1p1 |
| | FGOALS-g2 | r1i1p1 | | **MPI-GE\*** | **r(1-100)i1p3\*** |
| | *FIO-ESM* | *r(1-3)i1p1* | MRI | MRI-CGCM3 | r1i1p1 |
| NOAA GFDL | GFDL-CM3 | r1i1p1 | | MRI-ESM1 | r1i1p1 |
| | GFDL-ESM2G | r1i1p1 | NCC | NorESM1-M | r1i1p1 |
| | GFDL-ESM2M | r1i1p1 | | NorESM1-ME | r1i1p1 |
| NASA GISS | GISS-E2-H-CC | r1i1p1 | BCC | bcc-csm1-1-m | r1i1p1 |
| | *GISS-E2-H* | *r(1-2)i1p1* | | bcc-csm1-1 | r1i1p1 |
| | GISS-E2-H | r1i1p2 | | inmcm4 | r1i1p1 |
| | *GISS-E2-H* | *r(1-2)i1p3* | **Total** | | **288 members** |

**Figure 1.** Observational estimates (OBS; gray), the CMIP5 ensemble (blue), and the three SMILEs: CESM1.2.2-LE (red), CanESM2-LE (yellow), and MPI-GE (green) evaluated in this study, shown in terms of area- and seasonally-averaged absolute surface air temperature timeseries (SAT; °C). The two OBS datasets, ERA-20C Temperature and the Berkeley Earth Surface Temperature (BEST) product, are shown in solid gray and dashed gray respectively. Their average, used to determine member performance, is shown in solid black. For the CMIP5 and three SMILEs, the ensemble means across members are shown in solid color; the shading indicates the 5th-95th percentile of each distribution as a measure of ensemble spread. Note that the CMIP5 ensemble is a multi-model, multi-initial condition member ensemble of 88 members from 40 (named) model setups, not the "one model, one vote" ensemble often used in multi-model ensemble studies. Panel a shows projections for Northern European Winter (DJF NEU) and panel b shows projections for Mediterranean summer (JJA MED) SAT. The number of members in each ensemble is indicated in parenthesis in the legend.

elling center/known development streams that are grouped as dependent entities under the fourth independence assumption we

investigated.

The CESM1.2.2-LE used in this study was derived from a 4700-yr CESM control simulation with constant preindustrial forcing generated at ETH Zürich (Sippel et al., 2019). CESM1.2.2 uses the Community Atmosphere Model, version 5.3 (CAM5.3) and has a horizontal atmospheric resolution of 1.9° x 2.5° with 30 vertical levels (Hurrell et al., 2013). The preindustrial control run was branched at 20-year intervals, starting from the year 580, to create an ensemble with "macro" initial conditions,

i.e., different coupled initial conditions picked from well separated start dates (Stainforth et al., 2007; Hawkins et al., 2016). Members of the macro initial condition ensemble were run from 1850-1940 driven by historical CMIP5 forcing (Meinshausen et al., 2011). At year 1940, each macro initial condition member was branched into four different realizations, each subject to an atmospheric temperature perturbation of $10^{-13}$ to create "micro" initial condition ensembles (Hawkins et al., 2016). From these micro initial condition ensembles, 50 members were selected for the CESM1.2.2-LE (specifically, four micro ensemble

members from macro ensemble members 1 through 12 and two micro ensemble members from macro ensemble member 13).

The MPI-GE was generated using the low resolution set up of the MPI Earth System Model (MPI-ESM1.1) (Giorgetta et al., 2013). The 100 member ensemble has macro initial conditions: a preindustrial control simulation was branched on the first of January for selected years between 1874 and 3524 to sample different states of a stationary and volcano-free 1850 climate (Maher et al., 2019). The MPI-GE uses ECHAM6.3 run in a T63L47 configuration (Stevens et al., 2013) as its atmospheric component for a horizontal resolution of approximately 1.8°.

The CanESM2-LE (Arora et al., 2011) was initiated from the five CanESM2 members contributed to CMIP5 (which are included in our CMIP5 basis multi-model ensemble). As with CESM1.2.2, the CanESM2 large ensemble has a combination of macro and micro initial conditions. Macro initial conditions were taken from year 1950 of the five original CanESM2 members. Each were then branched 10 times with micro initial conditions (a random permutation to the seed used in the random number generator for cloud physics) to give a total of 50 members (Swart et al., 2018). The CanESM2-LE uses the CanAM4 atmosphere model run at a T63 spectral resolution.

The CMIP5 ensemble and three SMILEs are shown in terms of their respective ensemble means and spreads (represented by the 5th-95th percentile of each distribution) in Figure 1, for the two regions/seasons of interest: Northern European (NEU) winter (December-January-February; DJF) SAT (panel a) and Mediterranean (MED) summer (June-July-August; JJA) SAT (panel b). The NEU and MED regions used are the SREX regions defined in Seneviratne (2012). All models are forced with historical CMIP5 forcing from 1950-2005 followed by Representative Concentration Pathway 8.5 (RCP8.5) forcing from 2006–2099 (Meinshausen et al., 2011). The multi-model CMIP5 ensemble (Fig.1 blue) has a larger spread than the single model SMILEs, demonstrating that model uncertainty does rise above well-defined estimates of internal variability in the two European regions and seasons considered. The combined macro-micro perturbation CESM1.2.2-LE (Fig.1 red) has a larger ensemble spread than the CanESM2-LE (Fig.1 yellow), but, on average, warms less by end-of-century. The MPI-GE (Fig.1 green) has approximately the same amount of JJA MED warming as the CMIP5 ensemble average.

In addition to the multi-model ensemble, several observational estimates are used to assess model performance. Two global atmospheric reanalysis products, ERA-20C and NOAA-CIRES-DOE 20th Century Reanalysis V3 (NOAA-20C), represent observed SLP, while ERA-20C and a merged temperature data set, Berkeley Earth Surface Temperature (BEST), represent observed SAT. ERA-20C was created by European Centre for Medium-Range Weather Forecasts (ECMWF) and assimilates surface pressure and marine wind observations over the 20th century (1900-2010) into the IFS version Cy38r1 model (Poli et al., 2016). NOAA-20C, a co-effort between the National Oceanic and Atmospheric Administration (NOAA), the Cooperative Institute for Research in Environmental Sciences (CIRES), and the U.S. Department of Energy (DOE), assimilates surface pressure observations into the NCEP GFS v14.0.1 model to provide output from 1836-2015 (Compo et al., 2011; Slivinski et al., 2019). BEST was created to be an independent estimate of global temperature, obtained through spatiotemporal interpolation of in situ temperature measurements (Rohde et al., 2013).

The Knutti et al. (2017) weighting scheme can comprehensively account for observational uncertainty (Brunner et al., 2019), but for this study, we chose to use the average of two observational estimates in order to have a simple and straight-forward definition of climate within which the sensitivity of the weighting scheme can be interrogated. ERA-20C and NOAA-20C reanalyses were chosen because they provide temporally and spatially complete fields that extend back to 1950. Additionally,

as reanalysis products are, after all, model-based, we chose a reanalysis product with both SLP and SAT available (ERA-20C), as well as SAT and SLP fields from different sources (NOAA-20C and BEST). We further used the SLP-SAT relationship to obtain the circulation-induced component of SAT, which is removed to obtain the estimated residual thermodynamic SAT trends (see Appendix A). Though all products are observational estimates, we henceforth refer to them as "observations" or "OBS" to distinguish them from members of the multi-model ensemble.

## 3 Weighting Schemes

The weighting strategies used to constrain uncertainty in this study are rooted in a combination performance and independence weighting metric developed by Knutti et al. (2017), following on the work of Sanderson et al. (2015a, b). Summarized in the subsections below, the five strategies considered arise from common assumptions surrounding plausibility and similarity made about constituents of multi-model ensembles. With the exception of the first strategy, which assigns each member an equal weight, the basic principle of the weighting is as follows: a member will receive a performance weight based on how closely it resembles observed climate (based on nine chosen predictors; detailed in the following section). That performance weight will then be scaled by a measure of dependence that represents whether (or to what degree) a member is identified as a "duplicate" of another member over the historical period. It is important to note that dependence in this study is never determined by future behavior. Doing so would jeopardize the "agreement suggests robustness" paradigm by penalizing convergence. Rather, dependence is either a model property decided upon beforehand or determined through RMSE distances between historical aspects of climate.

### 3.1 Equal Weighting

The first way in which the multi-model ensemble is weighted is by all members receiving a weight, $w_i^I$, of 1.

$$w_i^I = 1 \tag{1}$$

This equal weighting follows from the assumption that all multi-model ensemble members are independent and equally plausible and is sometimes referred to as a "model democracy" assumption (Knutti et al., 2010a; Knutti, 2010). In instances where SMILEs are incorporated into a multi-model ensemble, the equal weighting strategy is clearly flawed; 50-100 members from the same model is a clear voting advantage within the model democracy. However, equal weighting serves as a baseline handling of multi-model ensemble information against which other weighting strategies can be compared.

### 3.2 Performance Weighting

The second weighting strategy builds upon the first in that all members are still assumed to be independent, but some members are identified to be more realistic than others. Members are thus weighted ($w_i^{II}$) by a measure of performance, here, based on the numerator of the Knutti et al. (2017) weighting function.

$$w_i^{II} = e^{-\frac{D_i^2}{\sigma_D^2}} \tag{2}$$

The term $D_i$ represents the RMSE distance between a multi-model ensemble member and observations; $w_i^{II}$ decreases exponentially as members increasingly differ from observations ($D_i >> 0$). A shape parameter $\sigma_D$ dictates the width of the performance weight Gaussian, determining how far apart a member and observations must be to be down-weighted. For a smaller value of $\sigma_D$, models are more rapidly down-weighted as they diverge from observed climate which often results in a weighting where few models receive weights of meaningful magnitude. For a larger value of $\sigma_D$, models are not as strongly penalized for not resembling observations which often results in a more even distribution of weights within the ensemble. Here, we select $\sigma_D$ to be 0.32 for the DJF NEU weighting and 0.4 for the JJA MED weighting (further discussion in Appendix B).

### 3.3 1/N scaling, IC members

The third weighting strategy extends the performance weighting by including a dependence assumption, making it suitable for the combined CMIP5-SMILE ensemble we evaluate. Each "model" gets a unique weight. The independent entity, "model", is assumed to be determinable by name (as listed in Table 1), which renders members of IC ensembles within the multi-model ensemble (the 13 within the CMIP5 ensemble and the three SMILEs) dependent entities. To achieve the model weighting, models that are represented by one member receive their performance weight $w_i^{II}$. Models that are represented by IC members receive an average of the performance weights of their N constituents ($\frac{1}{N}\Sigma_1^N e^{-\frac{D_j^2}{\sigma_D^2}}$). That average performance weight, divided by N, is assigned to each IC member. Therefore, the weight each member receives, $w_i^{III}$, is:

$$w_i^{III} = \frac{[\frac{1}{N_i}\Sigma_1^{N_i} e^{-\frac{D_j^2}{\sigma_D^2}}]}{N_i} \tag{3}$$

Each IC member is assigned the average performance weight of the IC ensemble (rather than its individually computed performance weight) to reflect the assumption that all IC members represent an equally likely outcome of the model. This choice rectifies the fact that when computed by RMSE, performance weights differ between IC members due to internal variability.

### 3.4 1/N scaling, modelling center

The fourth weighting strategy is identical to the third, but has a different definition of "model". The independent entity is determined not by name, but by a conjecture about model origin. Similar to the "same-center hypothesis" (Leduc et al., 2016), we group all members provided by a modelling center and/or in a known development stream (i.e., the CESM1.2.2-LE is grouped with the NCAR models, though it was run at ETH Zürich) as dependent entities. The weight of each model, $w_i^{IV}$, is computed as in the IC case, with averages taken over N, the number of members that constitute a model.

$$w_i^{IV} = \frac{[\frac{1}{N_i}\Sigma_1^{N_i} e^{-\frac{D_j^2}{\sigma_D^2}}]}{N_i} \tag{4}$$

## 3.5 RMSE distance scaling

Finally, the fifth weighting strategy operates under the assumption that dependence cannot necessarily be determined by model
name, but shared biases in simulating historical climate can give an idea of dependence that comes from differently named
models sharing ideas and code. Instead of relying on knowledge of model origin, the RMSE weighting $(w_i^V)$ initially proposed
by Knutti et al. (2017) relies solely on model output to determine a model's overall weight. It features an independence scaling
based on RMSE distance metrics in addition to the RMSE-derived performance weights. For results to be compatible with past
assessments of this weighting scheme (e.g. Lorenz et al., 2018; Brunner et al., 2019), we assign each member their unique
performance weight (as computed in $w_i^{II}$) even if they are IC ensemble members. This puts the RMSE weighting in contrast
to the 1/N scaling approaches which ensure IC ensemble members have identical weights.

$$w_i^V = \frac{e^{-\frac{D_i^2}{\sigma_D^2}}}{1 + \Sigma_{j \neq i}^M e^{-\frac{S_{ij}^2}{\sigma_S^2}}} \tag{5}$$

$S_{ij}$ represents the distance between multi-model ensemble member $i$ and multi-model ensemble member $j$. Unlike in the
1/N strategies, the RMSE independence scaling is based solely on $S_{ij}$, how far a member is from all the other members in the
ensemble, and not on any prior knowledge of the multi-model ensemble member's origin. As with the performance weight, a
shape parameter $\sigma_S$ dictates the width of the Gaussian that is applied to the member pairs. $\sigma_S$ represents how close a member
must be to another member before they are considered dependent entities. For a member with no close neighbors ($S_{ij} >> \sigma_S$),
the independence scaling tends to 1, preserving the member's overall weight. For a member with many close neighbors ($S_{ij} << \sigma_S$), the independence scaling is greater than 1 and reduces its overall weight. For the CMIP5-SMILE ensemble, the goal
is to select a $\sigma_S$ that is large enough such that members of a SMILE are considered dependent entities, but not so large that
the majority of multi-model ensemble members are considered dependent as well. Here, we select DJF NEU $\sigma_S$ to be 0.25 and
JJA MED $\sigma_S$ to be 0.26. Sensitivity to the choice of $\sigma_S$ and further details on selection strategies are discussed in Appendix B.
Upon computation of the weights in each strategy, each weight is normalized by $\Sigma_i w_i$ such that they sum to 1.

## 3.6 Defining "Climate": Predictor Selection

The performance weight used in weighting strategies two through five and the independence scaling used in strategy five are
based on a chosen definition of climate. A model's performance is based on its ability to reproduce observed climate. Under
assumption five, a member's independence is based on how much its climate differs from the climate in other members. When
defining climate, the aim is to optimize the "fitness for purpose", which should include choosing predictors that are physically
associated with the target and will indicate if a model is biased in a way that renders it unsuitable for realistic simulation of the
target. For example, in Knutti et al. (2017), aspects of climate relevant for September sea ice extent, such as the climatological
mean and trend in hemispheric mean September Arctic sea ice extent, were chosen. These chosen predictors reflected that
models with almost no sea ice in the present day or significantly more sea ice in the future than presently observed were

less suitable for the task of projecting changes in sea ice extent. It is also good practice to avoid using a single predictor to define climate to avoid an over-confident uncertainty estimate. No one model property can comprehensively reflect if the

265 model is "good" for a particular purpose, and it is dangerous to constrain uncertainty by dismissing models that do not match observations by a particular statistical definition for those that happen to be tuned to match that definition. Lorenz et al. (2018) discusses a more holistic strategy for choosing predictors and ultimately selected from a set of 24 predictors deemed relevant for projecting North American maximum temperature, based on known physical relationships, predictor-target correlations, and variance inflation considerations.

Here "fitness for purpose" is a relatively simple and straight-forward definition of climate within which the sensitivity of the weighting scheme can be interrogated. We base the performance weighting and the RMSE independence scaling on nine predictors: the climatology and interannual variability (represented by standard deviation) of SAT and SLP during the periods of 1950-1969 and 1990-2009 and a 50-year derived SAT trend (estimated residual thermodynamic trend; described in more detail in subsequent paragraphs) for the period of 1960-2009. We chose predictors to be aspects of regional temperature and

pressure in a domain that encompasses modes of atmospheric circulation variability relevant to European climate, because they are (1) physically associated with the target (end-of-century warming) and (2) fields that may reflect model biases that would affect realistic simulation of future climate. For example, a model with a warmer-than-observed mean state in the Mediterannean may experience an enhanced land-atmosphere feedback mechanism that amplifies drying and warming of the region (e.g. Christensen and Boberg, 2012; Mueller and Seneviratne, 2014; Vogel et al., 2018). SAT and SLP are found to

be highly relevant predictors by earlier studies (Brunner et al., 2019) and are among the most comprehensively measured atmospheric fields prior to the satellite era (Trenberth and Paolino, 1980). In terms of spatial domain, SAT climatology and variability predictors are computed over their corresponding ocean-masked SREX regions (i.e. NEU for DJF and MED for JJA) and SLP climatology and variability predictors are computed over a larger European sector domain that includes the North Atlantic ($25 - 90°$N and $60°$W$-100°$E). The derived SAT trend, estimated residual thermodynamic trend, is computed

over the ocean-masked continental European domain (EUR; $30 - 76.25°$N and $10°$W$-39°$E).

    To compute the aggregate distance metrics from nine predictors, all predictor and observational fields are bilinearly interpolated to a shared $2.5°$ x $2.5°$ latitude-longitude grid. The predictors are then time-aggregated, with the mean or standard deviation computed over the periods 1950-1969 and 1990-2009, and the estimated residual thermodynamic trend computed over the period 1960-2009. For each time-aggregated predictor, the differences between the observed mean value and member

value (or member value and member value in the case of the RMSE independence scaling) are computed at each grid point and subsequently squared. The squared differences are then area-averaged over the predictor domain and square-rooted to obtain an RMSE distance for observed-member and member-member pairs. For each predictor, the resulting distributions of observed-member and member-member RMSEs are then normalized by their mid-range value ([maximum + minimum]/2), such that the distance for each of the nine predictors are on the same order of magnitude and can be combined into a single $D_i$

(Figure B1) or $S_{ij}$ value (Figure B2) for each member.

    A final consideration in predictor selection is one of relationships between past and future predictor behavior. A model's performance weight is based on its ability to reproduce observed climate, and this methodological choice follows from the

concept of emergent constraints (e.g. Hall and Manabe, 1999; Allen and Ingram, 2002; Borodina and Knutti, 2017). The assumption is that if a model accurately represents an aspect of historical climate, it is likely to realistically represent relevant physical processes and therefore is likely to provide a reliable future projection. If a model is significantly biased with respect to observed climate, its future representation of climate may be cause for concern (Knutti et al., 2017), particularly when a statistical relationship between the historical and future climate feature of interest exists. In the absence of a strong statistical relationship, predictors serve to add degrees of difference between members that helps to ward against overconfident weighting.

Statistical relationships between historical and future climate can be obscured by internal variability, and the inclusion of SMILEs in a multi-model ensemble highlights the need to understand the role of internal variability on the chosen predictors. In particular, internal variability is shown to influence trends in regional SAT even on the 50-year predictor timescales we have selected (Deser et al., 2016). Because of this, a member may have a similar-to-observed SAT trend (and thus a higher performance weight) by chance, simply because it has similar-to-observed climate variability over the trend period (i.e. a similar set of El Niño and La Niña events or similar phasing of the Atlantic Multi-decadal Oscillation). Because internal variability is inherently random in temporal phase (Deser et al., 2012), a member's match to observations over one trend period does not guarantee a match in the future. This issue is demonstrated in Figure 2ai, which shows that there is no discernible relationship ($R^2 \sim 0$) between the DJF EUR SAT trend from 1960-2009 and from 2050-2099 in CMIP5 with (black line) or without (blue line) the SMILEs. Even the two observational estimates differ in European winter trend by more than a degree over 50 years. In summer, a season with less midlatitude climate variation, a relationship emerges between 1960-2009 and 2050-2099 European SAT trends. The linear relationship between past and future trend is reinforced by the SMILEs in a model mean sense, i.e., the three new models added to the CMIP5 ensemble support the relationship (Fig.2bi). It is not evident within the SMILEs themselves, which reflects that the relationship is due to model differences and not the behavior of individual IC members.

The removal of the estimated influence of internal atmospheric variability from regional SAT, however, provides an alternative performance metric on which observations and models can be compared. Using a method of dynamical adjustment (described in Appendix A and in further detail in (Deser et al., 2016)), we construct an estimate of the component of SAT variability induced by large-scale atmospheric circulation patterns, remove it from the SAT record, and obtain the estimated residual thermodynamic trend for 1960-2009 and 2050-2099. The estimated residual thermodynamic trend is an estimate of both the influence of surface processes (i.e., land-atmosphere interactions (Lehner et al., 2017; Merrifield et al., 2017)) and the influence of the radiative forcing, an influence often defined as the forced response. In the model world, the forced response of a field is often defined as the ensemble mean or average across multiple ensemble members. However, there is no observational equivalent to the ensemble mean; there is only one observed realization of climate. Therefore, we use the estimated residual thermodynamic trend as a predictor because it can be computed in the same manner through dynamical adjustment in both observations and each multi-model ensemble member.

Internal atmospheric variability serves to amplify both observed SAT trends in winter by approximately 0.6°C. Removing the influence of dynamics results an average observed estimated residual thermodynamic trend that falls centrally within the CMIP5 and SMILE distributions (Fig.2aii). In summer, dynamical adjustment also centers estimated residual thermodynamic trend and slightly reduces the difference between observational datasets (Fig.2bii). In terms of weighting, the shift of observed values

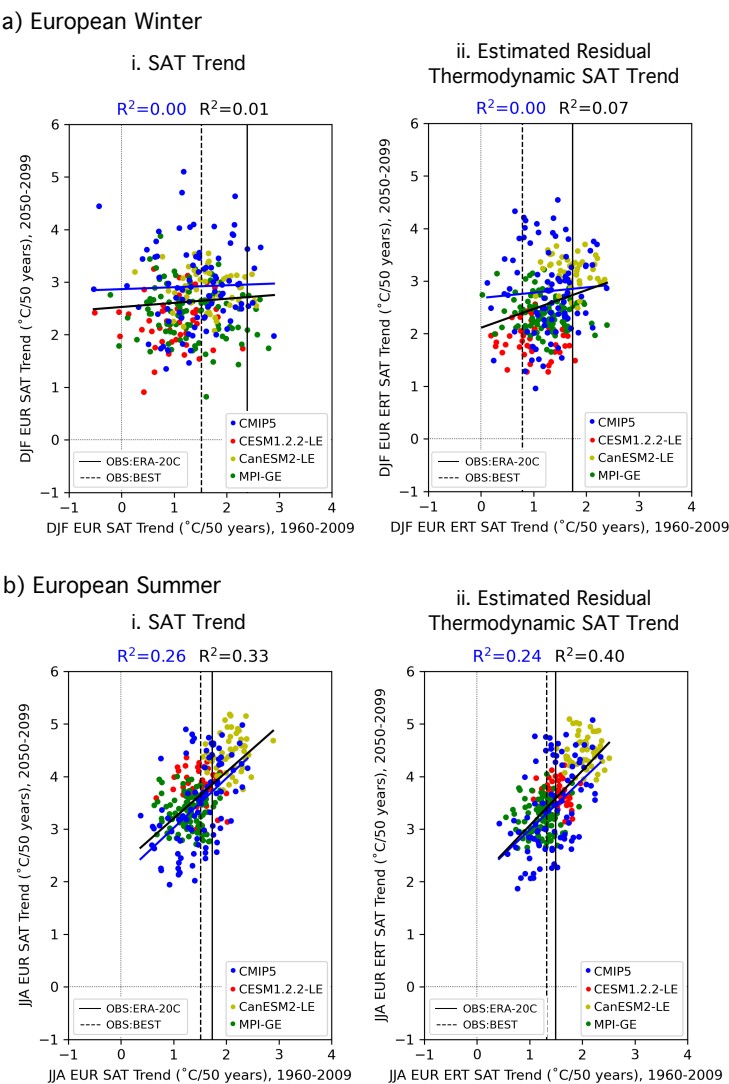

**Figure 2.** Predictor relationships of the domain-averaged 50-year trends of a) DJF and b) JJA European (EUR) SAT. 50-year raw trends are shown in panel i and 50-year estimated residual thermodynamic trends are shown in panel ii. In each panel, 1960-2009 is shown on the abscissa and 2050-2099 is shown on the ordinate. ERA-20C (BEST) observational estimates of the 1960-2009 trends are indicated by the solid (dashed) vertical lines. Least-squares regression fits (solid lines) and $R^2$ values computed for solely the CMIP5 output are shown in blue and computed for ALL output (CMIP5 and the three SMILEs) are shown in black.

to the center of the model distribution will lead to more models "performing" in their simulation of trend, which will, in turn, allow more models to contribute to the uncertainty estimate. The estimated residual thermodynamic trend can also be thought of as a property of each model, a measure that includes the response to the shared forcing, analogous to climate sensitivity (Knutti et al., 2017). We find that SMILE members, which share both model setup and forcing, also tend to have similar

estimated residual thermodynamic trends (Fig.2a,bii). In winter, the clustering of SMILE estimated residual thermodynamic trends is striking in comparison with SMILE trends: CESM1.2.2-LE members tend to have the least EUR warming in both periods, while CanESM2-LE members tend to warm the most. The addition of the SMILEs then introduces a slightly positive relationship between past and future responses (Fig. 2aii, black trend line) not apparent in the CMIP5 ensemble (Fig. 2aii, blue trend line), though no strong relationship emerges from variability in either case. In summer, the positive relationship seen between past and future Mediterranean SAT trends (Fig.2bi) is robust to the combination of removing internal atmospheric variability and adding the SMILES (Fig.2bii). CanESM2 has the most JJA MED warming in both the past and future periods, while MPI-GE has the least. Because estimated residual thermodynamic SAT trends in the broader European region are more comparable between members and observations due to the removal of an estimate of the influence of atmospheric variability that manifests on multi-decadal time-scales, we chose them as the ninth predictor in the definition of climate used in our performance weightings and RMSE independence weighting. Emergent relationships within the other eight predictors are discussed in Appendix C.

## 4   Results

To assess the influence of the weightings, we evaluate the magnitude of regional European end-of-century warming in terms of the SAT change ($\Delta$) from 1990-2009 climatology to 2080-2099 climatology. Two ensembles are considered, one comprised solely of CMIP5 members (CMIP5; distribution of 88 values) and one comprised of all available members from CMIP5 and the three SMILEs (ALL; distribution of 288 values). The CMIP5 and ALL SAT $\Delta$ distributions are shown side-by-side as box-and-whiskers elements in Figure 3 a and b for the five weighting strategies considered: equal, performance, 1/N scaling of IC members, 1/N scaling of modelling center contributions, and RMSE distance scaling. Weighted ensemble mean values are shown by solid horizontal lines within the box elements. Weighted ensemble spread is illustrated by the box, which indicates the 25th and 75th percentiles, and the whisker, which indicates the 5th and 95th percentiles.

For each weighting strategy, comparisons between the CMIP5 and ALL distributions help to elucidate (i) how the weighting constrains uncertainty in the magnitude of end-of-century regional European warming and (ii) how the inclusion of SMILE members influences the distribution. To explicitly determine the contribution of the SMILEs, we also show the fraction of total weight received by each SMILE and CMIP5 in Fig.3 c and d. Contributions are determined by summing the normalized weights of the 50 CESM1.2.2-LE members (red bar), 50 CanESM2-LE members (yellow bar), 100 MPI-GE members (green bar), and the remaining 88 CMIP5 members (blue bar).

For the most part, the weighting strategies introduce only modest distributional shifts; both Northern European winters and Mediterranean summers are projected to warm, most likely by about 5-6°C, by end-of-century (Fig.3a,b). What is more at issue than the distributional statistics, though, is what the distribution actually represents. An equal weighting results in a distribution representative of warming in the models with the most votes; in this case the SMILEs. In both seasons, the equal weighting demonstrates why it is important to treat SMILE members as dependent entities within a multi-model ensemble. The CMIP5 ensemble projects an ensemble mean end-of-century warming of 5.9°C and an interquartile spread of 2.2°C for Northern

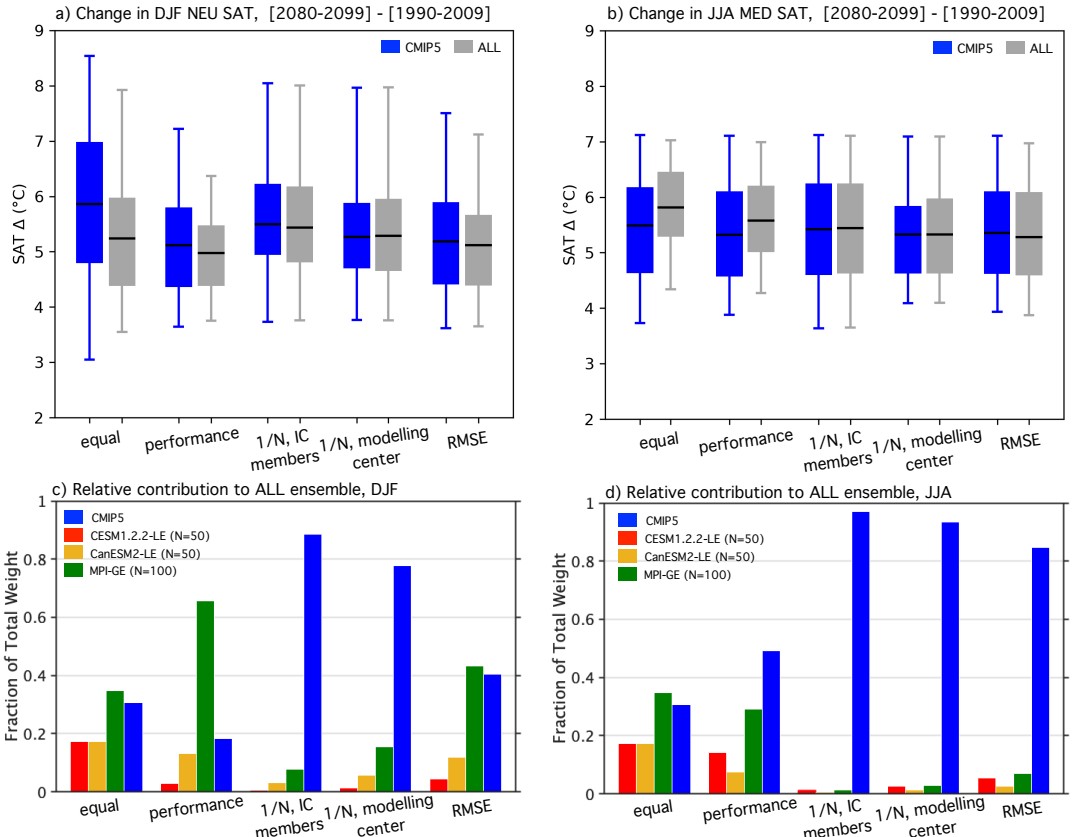

**Figure 3.** (a) Box-and-whisker plot showing how the five weighting strategies effect the distributions of DJF NEU SAT change ($\Delta$, [2080-2099]-[1990-2009]) for the CMIP5 ensemble (blue) and ALL ensemble (CMIP5 with the three SMILEs; gray). The box element spans the 25th to 75th percentile of the distribution; mean SAT change is indicated by the horizontal line within the box. The whisker element spans the 5th to 95th percentile. b) As in a), but for JJA MED SAT change. c) The contribution of SMILE and CMIP5 members to the DJF NEU ALL ensemble under different weighting strategies, in terms of fraction of total weight. d) As in c), but for the JJA MED ALL ensemble.

European winter (Fig.3a), and an ensemble mean end-of-century warming of 5.5°C and an interquartile spread of 1.5°C for Mediterranean summer (Fig.3b). The addition of 200 SMILE members to the 88 member CMIP5 ensemble shift the end-of-century warming distributions towards less DJF NEU end-of-century warming and more JJA MED end-of-century warming, and reduces interquartile spread by approximately 25% in both cases. The large contributions of the three added SMILEs artificially constrain uncertainty: the CESM1.2.2-LE and CanESM2-LE each receive 17.4% of the total ALL ensemble weight,

while the MPI-GE makes up the majority 34.7% (Fig.3c,d).

A performance weighting results in a distribution representative of warming in the models that historically get things right. By diminishing the contribution of members that differ from observational estimates, the performance weighting acts to constrain uncertainty in both the CMIP5 and the ALL ensemble. For DJF NEU SAT change, the performance weighting shifts

the CMIP5 ensemble mean downwards by $0.75°C$, the 75th percentile downwards by $1.2°C$, and 25th percentile downwards by $0.44°C$. This distributional shift towards less end-of-century warming is a due, in part, to members with SAT $\Delta$ greater than $8°C$ receiving weights that are two orders of magnitude smaller than the average assigned weight. Uncertainty in the DJF NEU ALL ensemble is constrained both by the performance weighting diminishing the contribution of CMIP5 members and because MPI is one of the highest performing models based on the chosen DJF predictors. The high performing MPI-GE receives $65.8\%$ of the total ALL ensemble weight, though individual MPI-GE members only receive up to three times more weight than the averaged assigned weight. The aggregate impact of 100 high performing members is outsized and results in the narrowing of the performance weighted end-of-century warming distribution. The narrowing does not reflect the increased certainty that comes from the agreement of independent entities within the ensemble. Instead, it exemplifies that there is a need for a dependence assumption in order to avoid the outsized influence that comes from being both historically realistic and numerously represented in the ensemble. For JJA MED SAT change, the performance weight reduces the contribution of the three SMILEs to the ALL distribution in comparison to the equal weighting case, with the largest reduction made to CanESM2-LE contribution ($17.4\%$ to $7.4\%$; Fig.3d). However, the three SMILEs (three independent entities) still receive $51\%$ of the total JJA MED ALL ensemble weight, their contributions again augmented by numerous representations. As in the equal weighting case, the JJA MED ALL performance-weighted ensemble mean is still modestly shifted towards more end-of-century warming than its JJA MED CMIP5 counterpart. This reflects the above CMIP5-average SAT change of the CESM1.2.2-LE and the CanESM2-LE in Mediterranean summer.

In an effort to more appropriately handle the mix of models and IC members present in the ALL ensemble, we next explore three scalings that reflect different member dependence assumptions: that IC members are dependent (Fig.3a,b; 1/N, IC members), that modelling center contributions are dependent (Fig.3a,b; 1/N, modelling center), and that members with similar historical climate are dependent (Fig.3a,b; 1/N, RMSE). The 1/N IC member scaling is based on the widely accepted assumption that IC ensemble members are, by definition, dependent. Originating from the same model setup, differences in IC members are not due to differences in model skill. Therefore, it follows that IC members should all receive the same performance weight which, in aggregate, reflects the skill of its basis model. We achieve this by averaging the performance weights of all members of a SMILE or CMIP5-based IC ensemble (Table 1, italic) and subsequently dividing this average performance weight by the number of members (N). This reduces the number of unique weights in the CMIP5 ensemble from 88 (each member receive a unique weight) to 44 and the number of unique weights in the ALL ensemble from 288 to 47.

The scaling of IC ensemble member weight within the CMIP5 ensemble (blue element) decreases DJF NEU end-of-century warming uncertainty and slightly increases JJA MED end-of-century warming uncertainty with respect to equal weighting. It is therefore evident that the IC ensembles within CMIP5, which range from 2 to 10 members, exert influence on the performance weighted DJF NEU distribution in the same way the SMILEs influence the corresponding performance weighted ALL distribution. While this is not seen in the corresponding JJA MED CMIP5 equal and performance weightings, it is important to note that even two or three extra votes for a high performing model are enough to influence uncertainty. The reduction of IC member influence is even more striking in the ALL distribution; the three SMILEs contribute $11.4\%$ of the total weight in the DJF NEU and $3.1\%$ in the JJA MED, down from performance weight contributions of $81.6\%$ and $50.7\%$ respectively. As with

other strategies, the 1/N IC member scaled DJF NEU ALL distribution is shifted towards less end-of-century warming with respect to its CMIP5 counterpart. The ALL and CMIP5 1/N IC member scaled JJA MED distributions are almost identical.

In addition to IC members, it is reasonable to assume that members of the same model that differ in resolution (i.e., MPI-ESM-LR and MPI-ESM-MR) or in component module used (i.e., MIROC-ESM and MIROC-ESM-CHEM) are dependent entities. However, determining where to draw the line between dependence and independence is difficult; models from different modelling centers share components, while models in a modelling center's development chain can differ from each other in most major parameterizations (Knutti et al., 2013). Here, we chose to take a logical approach to the dependent entity grouping, based largely on model name or knowledge of institution of origin (Table 1, Group). 1/N modelling center weights are computed in the same manner as the 1/N IC member weight within these broader groupings. The number of unique weights becomes 20 in both the CMIP5 and ALL ensembles because the CESM1.2.2-LE is grouped with the other NCAR models, the CanESM2-LE is grouped with the five members of CanESM2 in the CMIP5 ensemble, and the MPI-GE is grouped with MPI-ESM-LR and MPI-ESM-MR members. The 1/N modelling center scaling results in similar CMIP5 and ALL end-of-century warming distributions in both the DJF NEU and JJA MED, with distributions characterized by positive skewness and a narrower interquartile range than in the 1/N IC member scaling case. The SMILE contributions all approximately double from their 1/N IC member scaling levels, to contribute a combined 22.3% to the DJF NEU and 6.7% to the JJA MED ALL distributions respectively.

Finally, in the instance that dependence is not known *a priori*, an RMSE-based metric can be used to assign dependence. The idea is that because of model biases, dependent entities can be identified by their similar climates. Using the same set of predictors as used for performance, each member receives a unique weight: RMSE-based performance scaled by RMSE-based dependence. The RMSE independence scaling allows for more SMILE contribution than the 1/N independence scaling approaches (Fig.3 c,d) because internal variability distinguishes SMILE members from one another and thus allows them to be treated as separate entities. With more entities in the ensemble, it follows that the degree of dependence of the existing CMIP5 models increases (CMIP5 models become more dependent) in tandem with SMILE members degree of dependence decreasing (SMILE members become less-than-fully dependent). In the DJF NEU, it is striking that the high performing MPI-GE again contributes over 40% of the total weight. In the JJA MED, the RMSE independence scaling leads to comparable CMIP5 and ALL distributions with the ALL distribution projecting slightly less warming than the CMIP5 distribution. This is in contrast to the performance weighted case where the ALL distribution is narrower than and features more warming than the CMIP5 distribution.

## 4.1 Reconciling the RMSE and 1/N scalings

For the weighting approach introduced by Knutti et al. (2017) to be suitable for incorporating large initial condition ensembles into a multi-model ensemble, there must be a demonstrable reconciliation between the 1/N IC member and the RMSE independence scalings. The RMSE independence scaling has the ability to assign a degree of independence to all members. This addresses the issue that we may not truly know how independent a model is based on name or modelling center of origin alone. However, when dependent entities (i.e., SMILE members) are known, the RMSE metric must be able to identify them as dependent and scale their influence appropriately. In practice, this means we seek an RMSE scaling that approaches (or

exceeds) 1/N for the SMILEs and the IC ensembles within the CMIP5 ensemble. The goal of an RMSE scaling proportional to ensemble size comes with the understanding that scaling may be larger if the IC ensemble is very similar to other models or smaller if the IC ensemble is not fully identified as one model (as was the case with the nine predictor RMSE scaling).

One way to achieve an RMSE scaling that identifies IC members as dependent is to remove internal variability from the metric through predictor choice. While it would not be good practice to base member performance on few predictors because of over-confidence concerns, member dependence may be more accurately reflected by fewer predictors that distinguish models from one another. Advantages of choosing different sets of predictors for determining dependence and performance is two-fold: first, by selecting for ability to distinguish models rather than realism, dependence predictors can achieve a more substantial separation between SMILE-SMILE distances and SMILE-model distances. This reduces reliance on and sensitivity to the independence shape parameter $\sigma_s$ (Appendix B). Second, the "convergence to reality" paradox is no longer an issue; models will not be down-weighted for moving closer to observations (and thus each other) based on performance predictors.

We find that large-scale, long-term climatological averages are the most suitable predictors for this purpose because, in general, the influence of internal variability increases on smaller spatial-scales and shorter time-scales (Hawkins and Sutton, 2009; Lehner et al., 2020). The climatological time aggregation (CLIM) was chosen because, of the nine original predictors utilized, 20-year climatological averages cluster in SMILEs more than 20-year variability or 50-year derived trend values on regional scales (Figures 2, C1, and C2). We average over the entire historical period, 1950-2009, to obtain two long-term CLIM predictors: annually-averaged Northern Hemisphere SLP and annually-averaged global land SAT. The Northern Hemisphere region was selected for SLP to maintain the distinguishing characteristics of mean circulation biases in the target-relevant European sector (Figures C3,C4), while global land was selected for SAT to avoid convergence associated with models having similar average ocean temperatures. The RMSE independence scalings derived from the nine predictors in DJF and JJA are shown alongside the scalings derived from global land SAT and Northern Hemisphere SLP climatology predictors in Figure 4. RMSE independence scalings for each member of the ALL ensemble are indicated by thin horizontal colored line within their respective modelling center groupings. For comparison, the RMSE independence scalings are superposed on the 1/N modelling center scaling (gray bar).

In contrast to the nine predictor RMSE scaling (Figure 4a,b), the global land SAT-Northern Hemisphere SLP RMSE scaling allows for SMILE members to distinguish themselves and to approach or exceed 1/N values (Fig. 4c). In both DJF and JJA, no member of the ALL ensemble has a nine predictor RMSE scaling that exceeds 1/45. Inter-member RMSE distances, shown in panels a and b of Fig.B2, reflect why this occurs; SMILE members can be as different from one another as CMIP5 models are from each other. The nine predictor independence scaling is better able to distinguish SMILE members from CMIP5 members in JJA than in DJF (Fig.4b). With the global land SAT and Northern Hemisphere SLP CLIM predictors, SMILE members are clearly closer to one another than to other models, with the exception of the CanESM2-LE. Because the CanESM2-LE is created using the five CanESM2 contributions to CMIP5, the SMILE and CMIP5 contributions cluster as a 55-member CanESM2 ensemble within the ALL ensemble (Fig.4c). In terms of scaling, 55 CanESM2 members are scaled by an average of 1/55.0, while the CESM1.2.2-LE and the MPI-GE are scaled by an average of 1/48.7 and 1/100.5 respectively (Fig.4c). In addition to the SMILEs, other IC ensembles within CMIP5, such as the 10-member CSIRO-Mk3-6-0 ensemble, also achieve

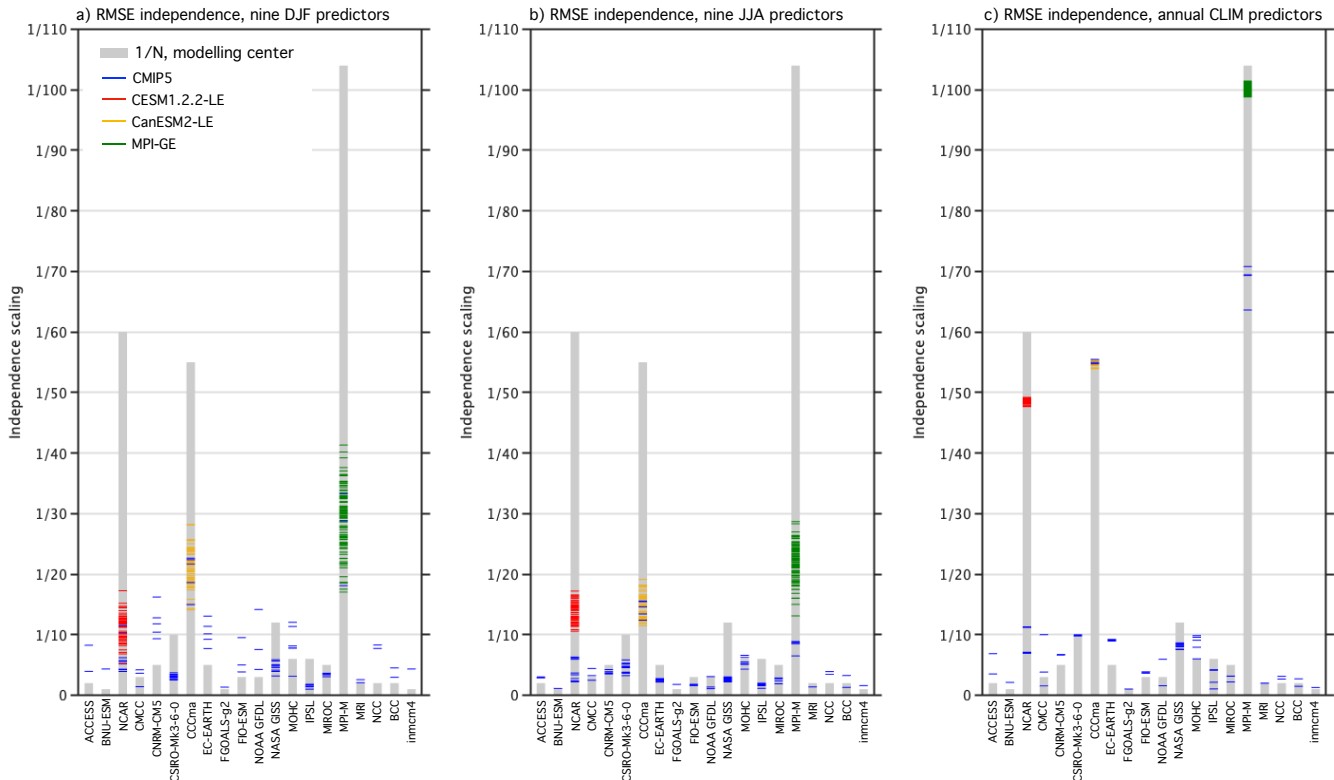

**Figure 4.** RMSE independence scalings (colored lines) of the SMILEs and CMIP5 ensemble members, grouped as listed in Table 1. CESM1.2.2-LE members are shown in red, CanESM2-LE members are shown in yellow, and MPI-GE members are shown in green, while the remainder of CMIP5 members are shown in blue. The 1/N modelling center scaling is shown in gray bars behind each grouping as a point of reference. Panels a and b show the scalings computed from the nine predictors used in the original DJF and JJA RMSE distance weightings respectively. Panel c shows the scalings computed from global land SAT and Northern Hemisphere SLP climatology predictors.

a 1/N scaling. Individually represented models, such as FGOALS-g2, are considered more independent and are thus scaled by factors that approach unity. On the other end of the dependence continuum, the four MPI-M contributions to CMIP5 are identified to have a high degree of similiarity to the MPI-GE and are scaled accordingly by factors exceeding 1/60.

To understand why large-scale, long-term CLIM predictors are able to group SMILE members and set a degree of dependence for CMIP5 members, we investigate where each member falls in the global land SAT and Northern Hemisphere SLP climatology predictor space in Figure 5. Each member is labelled either by color (SMILEs) or by model name (CMIP5) and IC ensembles within CMIP5 are circled. Circling IC ensembles within CMIP5 is possible because, along with the SMILEs, the IC members also tend to cluster. This phenomenon is in line with the assumption that IC members are dependent entities; the two large-scale, long-term CLIM predictors reflect this dependence. Notable IC clusters include MIROC5 (3 members) and EC-EARTH (5 members). The bifurcation in GISS-E2-H and GISS-E2-R ensembles reflect the p3 (top) vs. the p1 and p2

(bottom) perturbations used for different members. CanESM2 CMIP5 members join the CanESM2-LE (as indicated by Fig. 4c) and the MPI-ESM-LR contributions fall near the MPI-GE.

The assumption that members from the same modelling center are dependent entities, however, is not as clear cut in the global land SAT and Northern Hemisphere SLP climatology predictor space. GISS contributions share a response (lower Northern Hemisphere average SLP and higher global land average SAT), while the contributions from CMCC, GFDL, and IPSL feature markedly different responses (Fig.5). Another clustering feature present is that of several separate clusters for a modelling center. This can be seen for the NCAR modelling center grouping: CCSM4 and CESM1-BGC form a cluster

separate from both the CESM1-CAM5 cluster and the CESM1.2.2-LE cluster. The NCAR case illustrates that new models in a modelling center's development stream can be distinct from their predecessors and should not necessarily be considered dependent based on their shared name. On the other hand, there are also instances where models of different names are similar to each other. Bcc-csm1-1 falls within the CCSM4-CESM1-BGC cluster (Fig.5), which suggests that with shared components (Knutti, 2010), models can have similar responses and be identified as more dependent than their name would

suggest. Ultimately, discrepancies between model name and model response suggest that assigning each member a degree of dependence is a useful way to handle the continuum of dependence assumptions. Provided care is taken to select an appropriate set of predictors for independence scaling, IC members cluster in an anticipatable way while an interplay between named and unnamed model dependence remains.

## 5   Conclusions

We find that the performance and independence weighting scheme pioneered by Knutti et al. (2017) can be used to incorporate regional climate information from three single member initial condition large ensembles into a CMIP5 multi-model ensemble and return a justifiably constrained estimate of European regional end-of-century warming uncertainty. The performance weighting, which accounts for an ensemble member's ability to reproduce selected aspects of observed climate, is based on regional surface air temperature and sea level pressure climatology and interannual variability over two 20-year intervals during

the historical period (1950-1969 and 1990-2009) and a 50-year estimated residual thermodynamic SAT trend computed using a method of dynamical adjustment (Deser et al., 2016). These predictors bring both emergent relationships between past and future climate and aspects of climate that are important for a model to historically simulate in order to realistically project future warming to the definition of performance. The principle of emergent constraints underpins the choice to use estimated residual thermodynamic SAT trend over SAT trend, as the former is an estimate of a model-specific property that can be compared with

observations and the latter is influenced by internal variability even on 50-year timescales.

    Five different strategies based on the Knutti et al. (2017) performance and independence weighting are assessed for suitability of use in a CMIP5 and a combined CMIP5-SMILE ALL ensemble. While the different strategies introduce only modest distributional shifts (towards less end-of-century warming than in the equal weighting case), they imbue different meaning to the distribution. SAT change between 1990-2009 and 2080-2099 is projected to be about 5-6°C in both Northern European

winter and Mediterranean summer when historical model performance is considered. Equal and performance weighted ALL

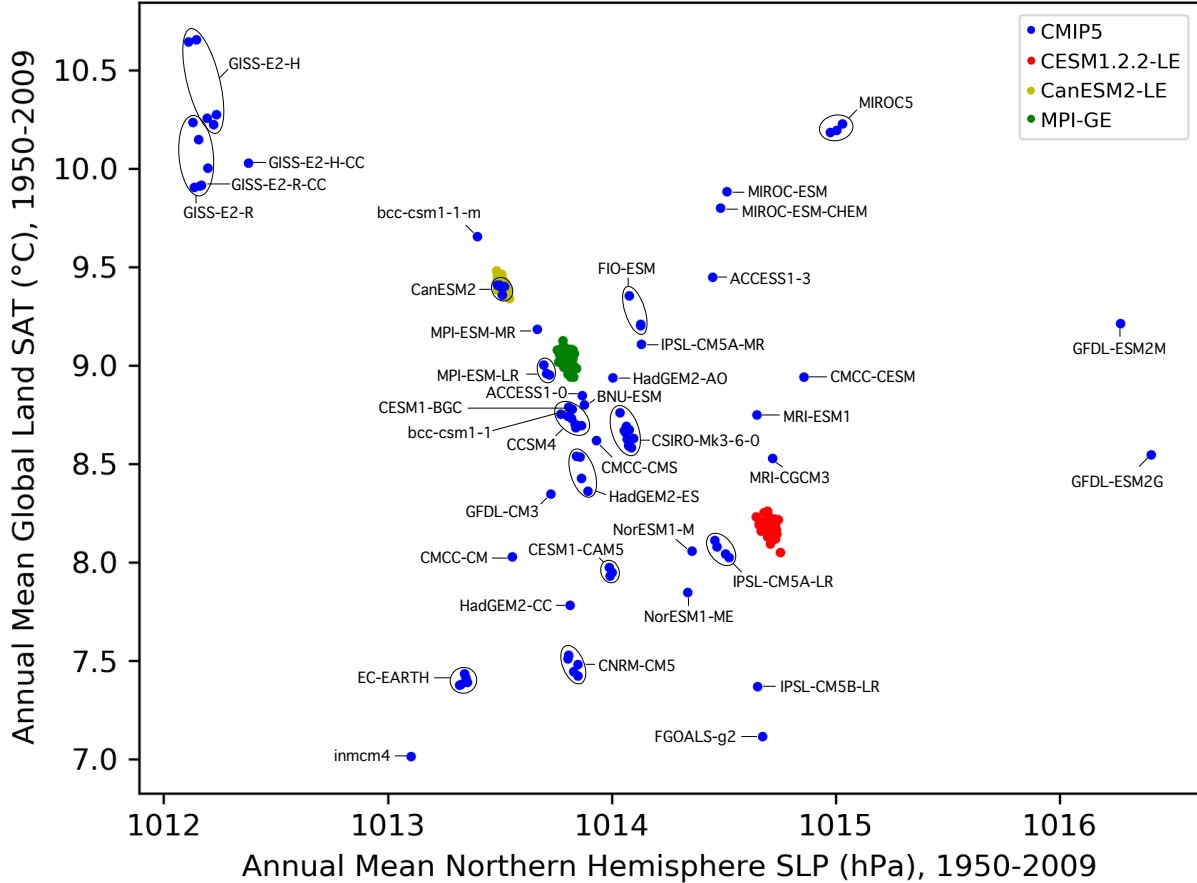

**Figure 5.** Scatter plot showing how ALL ensemble members distribute in the Northern Hemisphere SLP climatology / global land SAT climatology predictor space. Members and IC ensembles within the CMIP5 ensemble (blue) are labelled by model name. The CESM1.2.2-LE is indicated in red, the CanESM2-LE is indicated in yellow, and the MPI-GE is indicated in green, consistent with other figures.

distributions are narrowed by a 50-82% contribution from the SMILEs, which is an outsize contribution from three models to an ensemble comprised of 40 uniquely named models. The high performing, numerously represented MPI-GE receives over 65% of the total weight in the performance-weighted DJF NEU end-of-century warming distribution, demonstrating that an independence scaling is necessary so no one model defines the uncertainty range of a multi-model ensemble, regardless of its
historical realism.

Three plausible dependence assumptions are made to account for model contribution issues in an ensemble comprised of both known (i.e., IC members) and unknown (i.e., model component sharing) dependencies. By explicitly defining IC members as dependent entities, SMILE contributions drop to less than 10% while maintaining distributional shift tendency towards less end-of-century warming. Taking the definition of dependence a step further by considering all members from the same
modelling center and/or development stream dependent introduces positive skewness and a narrower interquartile range to

the distributions now containing 20 uniquely weighted entities. Finally, by acknowledging dependencies may not always be clearly determinable *a priori*, the independence scaling based on inter-member RMSE distances from the same nine predictors used to determine performance allows for reasonable levels of SMILE contribution to Mediterranean summer end-of-century warming uncertainty. However, the high performing MPI-GE contributes approximately 40% of the total weight to the Northern European winter distribution as a result of predictor internal variability distinguishing SMILE members as independent models.

The advantages of the RMSE-based independence scaling, which include allowing for degrees of dependence, are subverted somewhat by the inability of performance predictors to distinguish known dependent entities (i.e., IC members) from (presumed) independent ones. To address this issue, we show that a set of two predictors, 60-year annual average global land SAT climatology and 60-year annual average Northern Hemisphere SLP climatology, is capable of rendering an RMSE scaling of 1/N for SMILE members while assigning a degree of dependence to the rest of CMIP5. A notable achievement for these large-scale, long-term predictors is their ability to identify the CanESM2 members from CMIP5 as being from the same model version as the CanESM2-LE and scale the 55-member ensemble accordingly. A deeper look into groupings in the global land SAT and Northern Hemisphere SLP climatology predictor space reveals clustering of IC ensembles within the CMIP5 ensemble in addition to the SMILEs. MPI-ESM-MR and MPI-ESM-LR contributions cluster near the MPI-GE, while the NCAR model group separates into three distinct clusters consistent with NCAR's model development over time. The interplay between model name and model response does exhibit some complexity; models from the same center (i.e., GFDL) can have markedly different responses and models from different centers (bcc-csm1-1 and CCSM4) can have similar responses. This suggests that assigning degrees of dependence is a useful way to represent the information in an ensemble of opportunity like CMIP5.

It is important to note that while the weighting has a relatively straightforward functional form, it requires application-specific sets of predictors and appropriate shape parameters. Strategies to select optimal shape parameters are discussed in Appendix B of this study, and we advise that emergent predictor relationships are explored, as in Appendix C, to provide justification for the performance metric. When defining model skill for performance, it is important to carefully consider whether predictors are relevant to a model's ability to project the future target realistically. Different targets, such as hydrological changes, may require predictors to capture a more complex set of physical processes. It is also important to assess RMSE distance to observations of known dependent entities such as SMILEs to ensure internal variability in the selected set of predictors does not assign them skill of different orders of magnitude. Because SMILE members had relatively similar RMSE distance to observations over the nine original predictor, we did not require members of the SMILE to have identical performance weights under the performance and RMSE case assumptions evaluated. We, however, do see the merit in fixing IC member performance to an ensemble average value to ensure model skill is appropriately assigned. We also recommend that different sets of predictors be used for determining performance weight and independence scaling to avoid down-weighting independent models with historical climate that converges to reality. Independence predictors should be fields with minimal internal variability, such large-scale, long-term averages, and ideally fields that model developers do not explicitly tune, such as absolute global temperature (Mauritsen et al., 2012; Hourdin et al., 2017).

We assess a relatively unconventional multi-model ensemble in this study, which is comprised of 200 members from three models and only 88 members from the remaining 40 named models. This is a deliberate choice made to test and improve the independence scaling, as determining best practices for representing uncertainty in a multi-model ensemble that includes initial condition ensemble members is necessary in advance of CMIP6. Modelling centers are slated to submit more ensemble members to the project than were submitted to CMIP5 (Eyring et al., 2016; Stouffer et al., 2017). For more conventional

multi-model ensembles that include just a few initial condition ensemble members amongst the models, results may be less sensitive to choices underpinning the independence scaling. When large ensembles are included, however, it becomes clear that an independence scaling that scales known dependencies appropriately (i.e., 1/N for IC ensemble members), such as the RMSE global predictor scaling presented here, is necessary. Such an independence scaling will be a useful tool with which to assess uncertainty in the combined multi-model, multi-initial condition ensemble member CMIP6 ensemble.

**Appendix A: Dynamical Adjustment**

To obtain estimated residual thermodynamic trend in SAT, a method of dynamical adjustment, based on constructed circulation analogues, is used (Deser et al., 2016; Lehner et al., 2017; Merrifield et al., 2017; Guo et al., 2019). Dynamical adjustment provides an empirically-derived estimate of the SAT trends induced by atmospheric circulation variability; removal of this circulation-driven component from a SAT record thus reveals an estimate of the SAT trend associated with thermodynamic

processes and radiative effects. Dynamical adjustment relies on the ability to reconstruct a monthly mean circulation field, which we represent with sea level pressure (SLP) as in Deser et al. (2016), from a large set of analogues. Here, SLP analogues are selected from 60 possible choices (from the period 1950-2010), excluding the target month, and the method is therefore referred to as the "leave-one-out" method of dynamical adjustment. SLP fields in SMILE members, CMIP5 ensemble members, and the observational estimates ERA-20C and NOAA-20C are constructed in this manner for target months in the 1950-2010

period. For model years 2011-2099, analogues are selected from the entire 1950-2010 period. No notable trends in SLP have been identified over this period in previous dynamical adjustment studies (Deser et al., 2012, 2016; Lehner et al., 2017).

It is important to acknowledge that because of the paucity of analogue choices in leave-one-out dynamical adjustment, the term "analogue" is a bit of a misnomer. The term evokes the idea of a match, though in practice, analogues may not closely resemble the target. For convenience, we will continue to refer to the months used in target SLP construction as "analogues",

but we do so with the understanding that target and analogue patterns may differ over the selection domain.

A month is determined to be an analogue of the target month if the Euclidean distance between target and analogue SLP is small. Euclidean distance is computed at each grid point and averaged over the European sector domain also used for SLP predictors in the nine predictor RMSE weighting (25-90°N, 60°W-100°E). This selection metric, therefore, does not require an analogue to match the target month spatially over the whole domain. This is necessary because, with 60 possible options,

it is statistically unlikely that a "perfect" analogue will exist for a particular target month. van den Dool (1994) found that it would take on the order of $10^{30}$ years to find two Northern hemisphere circulation patterns that match within observational

uncertainty. With this in mind, a smaller than hemispheric domain and an iterative averaging schemes are employed to make the most of "imperfect" analogues available (Wallace et al., 2012; Deser et al., 2014, 2016).

Once the Euclidean distances are determined, the 50 closest SLP analogues are chosen, and the iterative process of selecting
30 of 50 SLP analogues and optimally reconstructing a target SLP field $\mathbf{X}_h$ commences. The optimal reconstruction of the target SLP is mathematically equivalent to multivariate linear regression; each analogue is assigned a weight ($\beta$) such that a weighted linear combination of analogues produces a least-squares estimate of the target SLP. $\beta$ is computed through a singular value decomposition of a column vector matrix $\mathbf{X}_c$ containing the 30 selected analogues and can also be estimated using through a Moore-Penrose pseudoinverse:

$$\beta = [(\mathbf{X}_c^T \mathbf{X}_c)^{-1} \mathbf{X}_c^T] \mathbf{X}_h \qquad\qquad\qquad (A1)$$

The analogue weighting scheme ensures that analogues which are further from (closer to) the target, in a Euclidean distance sense, contribute less (more) to the constructed SLP field.

After the target SLP field is constructed, the $\beta$ values derived for each SLP analogue are applied to their corresponding monthly-averaged SAT fields. Prior to the application of weights, a quadratic trend representing anthropogenic warming is
removed from the SAT record at each point in space. The purpose of this detrending is so that months picked from the end of the record do not contribute higher SAT anomalies simply because of the anthropogenically forced warmer background climate, even if the SLP patterns are the same (Lehner et al. 2017). Detrending strategies are further discussed in Deser et al. (2016). The weighted, detrended SAT fields are then used to construct a dynamic SAT anomaly field for the target month. SLP, which is representative of low-level atmospheric circulation, and SAT are physically related; SLP-derived weights are applied
to SAT to empirically construct that relationship. Conceptually, dynamic SAT anomalies are those that would occur given the attendant circulation pattern. The second through fifth steps of dynamical adjustment (selection of 30 of 50 SLP analogues, optimal reconstruction of target SLP, and construction of dynamic SAT) are then repeated 100 times, following Lehner et al. (2017). The dynamic component of SAT in the target month is the average of the 100 constructions. It is then subtracted from SAT in the target month to find the residual thermodynamic component of SAT, used as an estimate of the regional SAT
response to surface processes and radiative forcing. The trend of the residual thermodynamic SAT component is used as a predictor in this study; the trend is computed at each land grid point in the predictor domain and subsequently area-averaged.

## Appendix B: Selecting $\sigma_D$ and $\sigma_S$

Determining the shape parameters $\sigma_D$ and $\sigma_S$ is an important step in the RMSE weighting process (Knutti et al., 2017). $\sigma_D$ can be set using a perfect model test, as described in Lorenz et al. (2018). Here, a simplified perfect model test is performed on
a 47 member ensemble, which includes only the first IC member from the SMILEs and each of the CMIP5 models ensembles (40 named models with an additional 4 members from GISS-E2-R and GISS-E2-H physics physics-version ensembles). This is done because having multiple IC members (or a SMILE) in the ensemble could bias the perfect model test, which is based on predicting one member using a weighted distribution of the rest. We use member 1 for each IC ensemble because, often, when

multiple IC members are available, the first member is selected (e.g. Liu et al., 2012; Karlsson and Svensson, 2013; Sillmann et al., 2013). During the perfect model test, each member is assumed to be the "truth" once, and a weighting is performed using the remaining members to predict the "true" SAT change. RMSE distances (based on nine predictors) are computed with respect to the truth for the remaining members and used in the performance weighting function $w_i^{II}$ described in section 3.2. The performance weights are computed for $\sigma_D$ values ranging between 0 and 2 (on 0.01 intervals). For each $\sigma_D$, the weighted mean SAT change is computed and compared to the "true" SAT change. The optimal $\sigma_D$ for each truth is chosen to be where the difference between the weighted mean SAT change and the true SAT change is minimized. In the few cases when the weighted mean exhibits asymptotic behavior with no clear minimum difference prior to $\sigma_D = 2$, the $\sigma_D$ value is selected at the point where the leveling off begins (as determined by the intersection between a threshold value and the weighted mean curve). For the nine predictor RMSE weightings, we set $\sigma_D$ values to the mean of the 47 optimal $\sigma_D$ values computed during the perfect model test. It is important to note that this choice is ultimately subjective and further parameter sensitivity testing is recommended in studies focused on model performance.

The RMSE distances between multi-model ensemble members and observations ($D_i$) are shown in Figure B1. Members of the ALL ensemble are plotted in ascending order with the position of SMILE member indicated in red for the CESM1.2.2-LE, in yellow for the CanESM2-LE, and in green for the MPI-GE. In winter (Fig.B1a), distances between CMIP5 members and observations are distributed in a positively skewed fashion with the mode of the distribution at approximately $D_i = 0.40$ with a tail of larger $D_i$ values. In contrast, CMIP5 distances in summer (Fig.B1b) are approximately normally distributed about a mean of $D_i = 0.85$. The addition of the SMILEs to the distribution contribute to both of these distributional tendencies. $\sigma_D$ is set to 0.32 in DJF and 0.4 in JJA in both the CMIP5 and ALL ensembles to eliminate a degree of freedom of the method. Members are more strongly weighted by performance in winter than in summer, due to the different distance distributions.

$\sigma_S$ can be determined using IC ensembles present in the multi-model ensemble, including SMILEs. The inclusion of SMILE members in a multi-model ensemble emphasizes the need for $\sigma_S$ to be carefully selected, as SMILEs add redundant information and the purpose of $\sigma_S$ is to reduce the influence of redundant information. However, not all information added by a SMILE is distinguishable from information in other models in the nine predictor case; inter-member distances in an initial condition ensemble can be as large as inter-model distances in the multi-model ensemble (Figure B2a,b). Checking inter-member vs. inter-model distances is an important first step in determining $\sigma_s$, too much overlap between the distributions can blur the line between known dependent entities (IC members) and likely independent entities (different models).

If $\sigma_S$ is too small or too large, there are implications for the nine predictor RMSE weighted ensemble mean and spread. This sensitivity to $\sigma_S$ is shown in in Figure B3. We assess the characteristics of the nine predictor RMSE weighted CMIP5 distributions (Fig.B3a,bi) and RMSE weighted ALL distributions (Fig.B3a,bii) for different values of $\sigma_S$, varying from 0.05 to 0.8.

For small $\sigma_S$, only members that are very close to each other in predictor space are considered dependent; most members of the multi-model ensemble will therefore be considered independent. In this case, the RMSE weighting tends toward the performance weighted approach. If $\sigma_S$ is set on the order of the largest inter-member distances in a SMILE ($\sigma_S \geq \sim 0.5$), few members of the multi-model ensemble will be considered independent from each other, despite coming from different models.

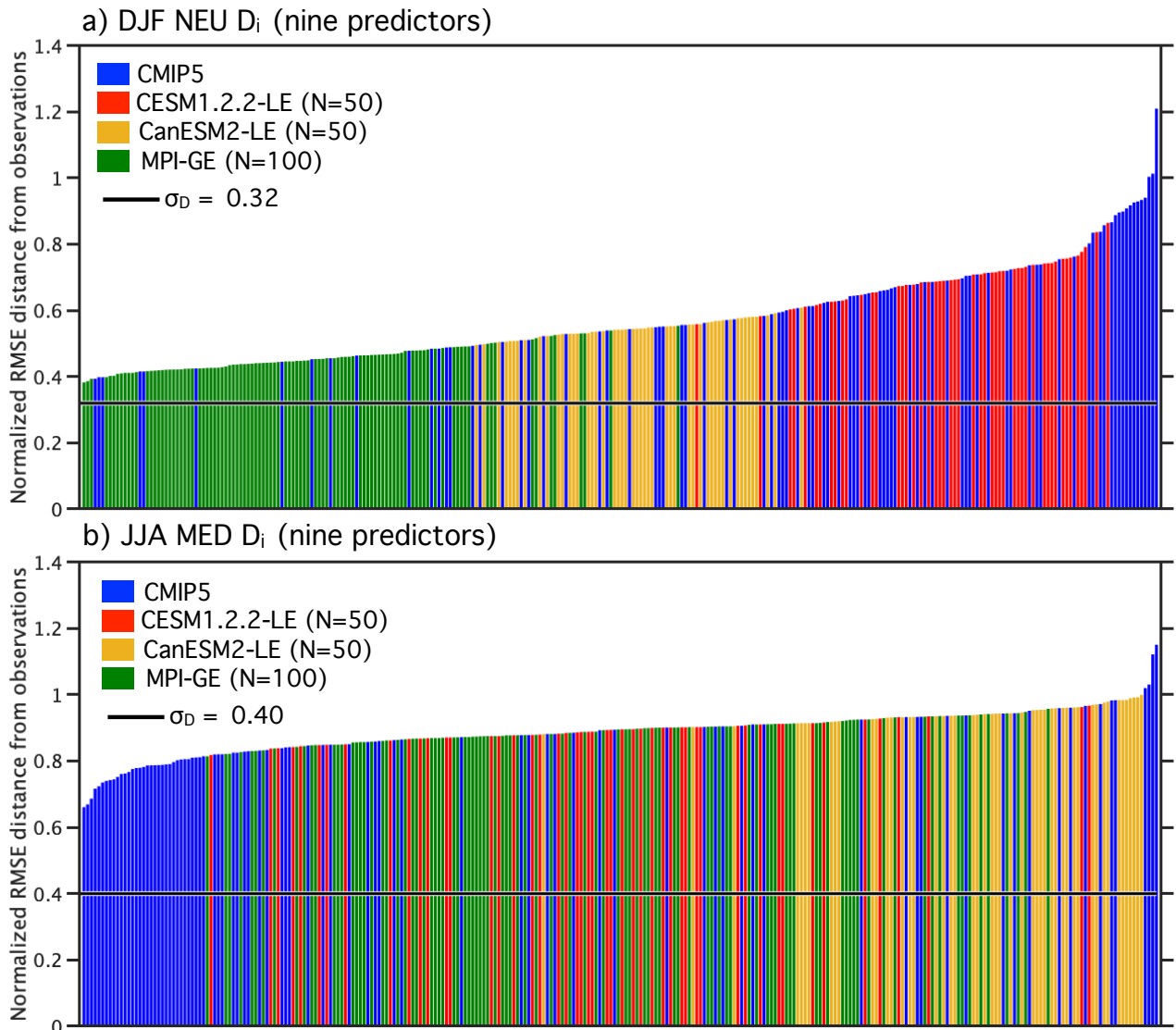

**Figure B1.** RMSE distance $D_i$, derived from nine predictors, between observations and the 288 members of the ALL ensemble (CMIP5 (blue) + CESM1.2.2-LE (red) + CanESM2-LE (yellow) + MPI-GE (green)). DJF NEU distances are shown in panel a and JJA MED distances are shown in panel b.

The systematic scaling of performance weights in the ensemble at large tends to also lead to a narrowing of uncertainty. Only members that are very far from other members will not have a scaled performance weight, but these "independent" members tend to also be far from observations and therefore have little performance weight to begin with. For $\sigma_S$ between approximately 0.2 to 0.4, uncertainty in the RMSE weighted distributions increases in all but the JJA MED CMIP5 case. The JJA MED CMIP5 distribution is relatively insensitive to $\sigma_S$ because 50% of the RMSE distances between CMIP5 members are between 0.56 and

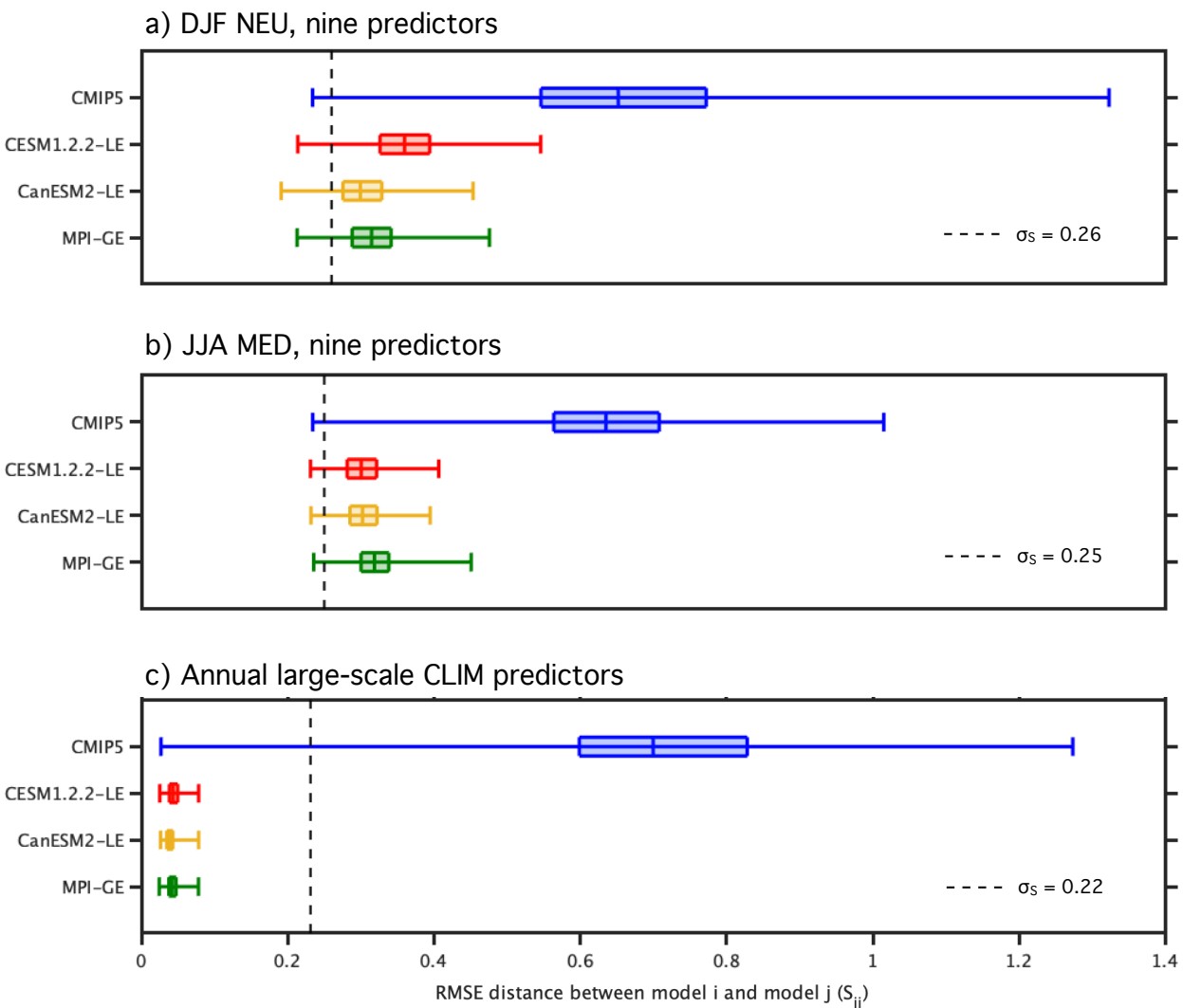

**Figure B2.** Distributions of RMSE distance (S$_{ij}$) within the SMILEs, the CESM1.2.2-LE (red), the CanESM2-LE (yellow), the MPI-GE (green) and the CMIP5 ensemble (blue). The box element spans the 25th to 75th percentile of the distribution; median S$_{ij}$ is indicated by the horizontal line within the box. The whisker element spans the full range of the S$_{ij}$ distribution. The value of $\sigma_S$ used for the weighting is indicated by the dashed line. DJF NEU distances based on the nine predictors are shown in panel a, JJA MED distances based on the nine predictors are shown in panel b, and distances based on annual global land SAT and Northern Hemisphere SLP climatology are shown in panel c.

0.71 (Fig.B2b). For the ALL distributions, the RMSE weighted mean shifts up modestly in DJF and down in JJA. In order to

avoid an underestimate of uncertainty, either due to redundancy or from down-weighting independent information, we propose that $\sigma_S$ should be set carefully. For the set of nine predictors, we set $\sigma_S$ based on the $S_{ij}$ distribution in IC ensembles present

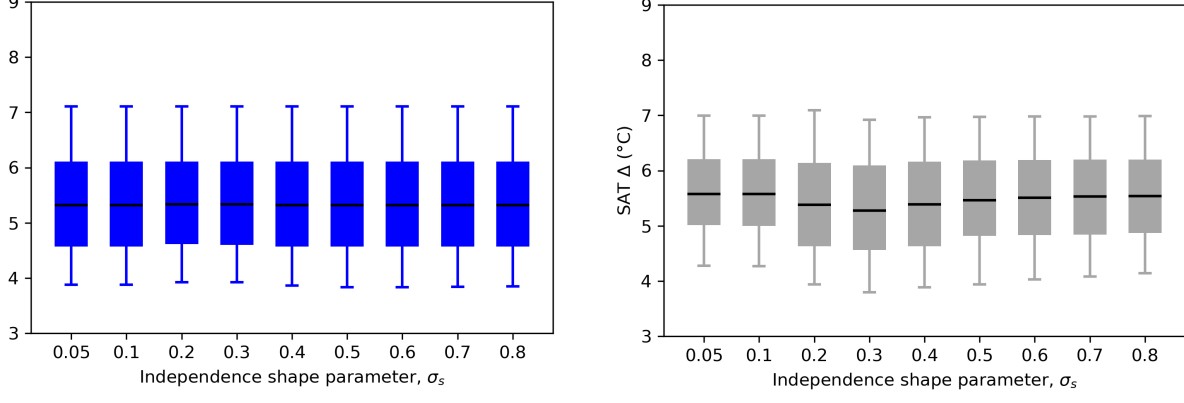

**Figure B3.** $\sigma_S$ sensitivity of the nine predictor RMSE weighting in Figure 3. $\sigma_S$ used for each weighting is indicated below each box element. Box-and-whiskers showing the SAT change distribution under the RMSE independence scaling weighting assumption ($\Delta$, [2080-2099]-[1990-2009]) for the CMIP5 ensemble (i; blue) and ALL ensemble (ii; gray). The box element spans the 25th to 75th percentile of the distribution; mean SAT change is indicated by the horizontal line within the box. The whisker element spans the 5th to 95th percentile. DJF NEU SAT change is shown in a and JJA MED SAT change is shown b.

within the multi-model ensemble. We compute the $S_{ij}$ within the three SMILEs and set $\sigma_S$ at two standard deviations below the SMILE $S_{ij}$ mean value (Fig. B2). The three values are then averaged. By this metric, DJF NEU $\sigma_S$ is 0.26 and JJA MED $\sigma_S$ is 0.25.

Another more robust option, as discussed in the main text, is to select a set of independence predictors that explicitly differentiate inter-IC member distances from inter-model distances. In this case, $\sigma_S$ should not be set to two standard deviations below the SMILE $S_{ij}$ mean, rather it should be set to a value greater than all IC member $S_{ij}$ but less than inter-model $S_{ij}$

(particularly differently named models). For the large-scale CLIM predictor set explored in Figure 4, $\sigma_S$ can be computed based on IC member intermember distances as described in Brunner et al. (2019); $\sigma_S$ in this instance is 0.22.

## Appendix C: Emergent Predictor Relationships


In addition to relationships between past and future (estimated residual thermodynamic) trend (Fig.2), emergent relationships among the remaining predictors we use to represent climate are shown in Figures C1 and C2. Linear relationships are clear for climatological averages in both seasons; multi-model ensemble member's climatological biases are more or less unchanged from past to future, with hotter mean state climate than other members during the historical period also tend to have hotter

mean state climate than other members in the future. Similarly, the tendency of domain-averaged SLP values to be and remain lower or higher also persists into the future. This relationship is explored spatially in Figures C3 and C4. Mean states within SMILEs tend to cluster together. With the exception of JJA MED SLP climatology (Fig.C2b), the addition of the SMILEs does not change the linear relationship found in the CMIP5 multi-model ensemble.

For variability (standard deviation over the given period), members of SMILEs differ as much from each other as from other

multi-model ensemble members in DJF (Fig.C1c,d). In JJA (Fig.C2c,d), several members of the CMIP5 multi-model ensemble have domain-averaged variability that falls outside the distribution of SMILE members. The addition of the SMILEs to the CMIP5 multi-model ensemble reduces correlations between historical and future variability for SAT and SLP in both seasons. This is particularly striking in JJA where the correlations tend to be due to the CMIP5 multi-model ensemble outliers.

Because the SLP predictor domain has a larger spatial extent than the SAT predictor domains, we also assess spatial patterns

of climatological SLP which average to the lowest and highest domain-averages values in the 1990-2009 climatological period (Figures C3 and C4). The "end-members" illustrate the climatological emergent constraint relationship seen in Figures C1 and C2 in terms of pattern, that is important for a field like SLP which tends to feature dipoles on basin and continental scales. For simplicity, we compare the end-members to one observational estimate from ERA-20C.

In winter, multi-model ensemble members tend to feature similar-to-observed spatial patterns of climatological SLP in the

predictor domain, with a low pressure center over the high latitude North Atlantic and a region of high pressure over the Eurasian continent (Fig.C3). For the member with the lowest domain-average, the difference arises from a further extension of the low pressure center across Northern Europe and a weaker high pressure center than observed, especially in the vicinity of the Tibetan plateau (Fig.C3 ii,v). For the member with the highest domain-average, the difference arises from high pressure features over high altitude regions, such as Greenland and the Tibetan plateau (Fig.C3 iii,vi).

In summer, members differ in spatial patterns of climatological SLP in the predictor domain, though most feature a high pressure center over the subtropical North Atlantic and lower pressure over the Eurasian continent seen in ERA-20C (Fig.C4). The member with the lowest domain-average features the afforementioned spatial pattern, but with a higher-than-observed amplitude i.e. both a higher North Atlantic subtropical high pressure center and a lower region of continental low pressure (Fig.C4 ii,v). In contrast, the member with the highest domain-average has high pressure over the entire Atlantic basin as well

as over Greenland and the Tibetan plateau (Fig.C4 iii,vi). Most importantly, in all cases, the climatological behavior of the past continues into in the future, which supports the primary tenet of an emergent constraint.

*Author contributions.* RK, RL, and LB conceived of and wrote the weighting scheme python package. ALM and LB implemented the weighting scheme with contributions from RL. ALM, LB, and IK analyzed the output. ALM wrote the paper with contributions from all co-authors.

*Competing interests.* We declare that we have no conflict of interest.

*Acknowledgements.* We would like to thank Drs. Nicola Maher, Flavio Lehner, Angeline Pendergrass, Sebastian Sippel, and two anonymous reviewers for their helpful comments on this manuscript. This project was funded by the European Union's Horizon 2020 research and innovation program under grant agreements 641816 (CRESCENDO) and 776613 (EUCP). We acknowledge the World Climate Research Programme's Working Group on Coupled Modelling, which is responsible for CMIP, and we thank the climate modeling groups for producing
and making available their model output. CMIP5 data was obtained from http://cmip-pcmdi.llnl.gov/cmip5/. The CESM1.2.2 large ensemble was generated at ETH Zürich and is available upon request. The CanESM2 large ensemble was generated by the Environment and Climate Change Canada's Canadian Centre for Climate Modelling and Analysis and is available at http://open.canada.ca/data/en/dataset/aa7b6823-fd1e-49ff-a6fb-68076a4a477c. The MPI Grand Ensemble was generated at the Max Planck Institute for Meteorology and is available esgf-data.dkrz.de/projects/mpi-ge/. ERA-20C data is provided by ECMWF and was obtained from https://apps.ecmwf.int/datasets/data/era20c-
moda/levtype=sfc/type=an/. The weighting protocol is available as a python package and can be obtained via GitHub (https://github.com /lukasbrunner/ClimWIP) under a GPLv3.

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

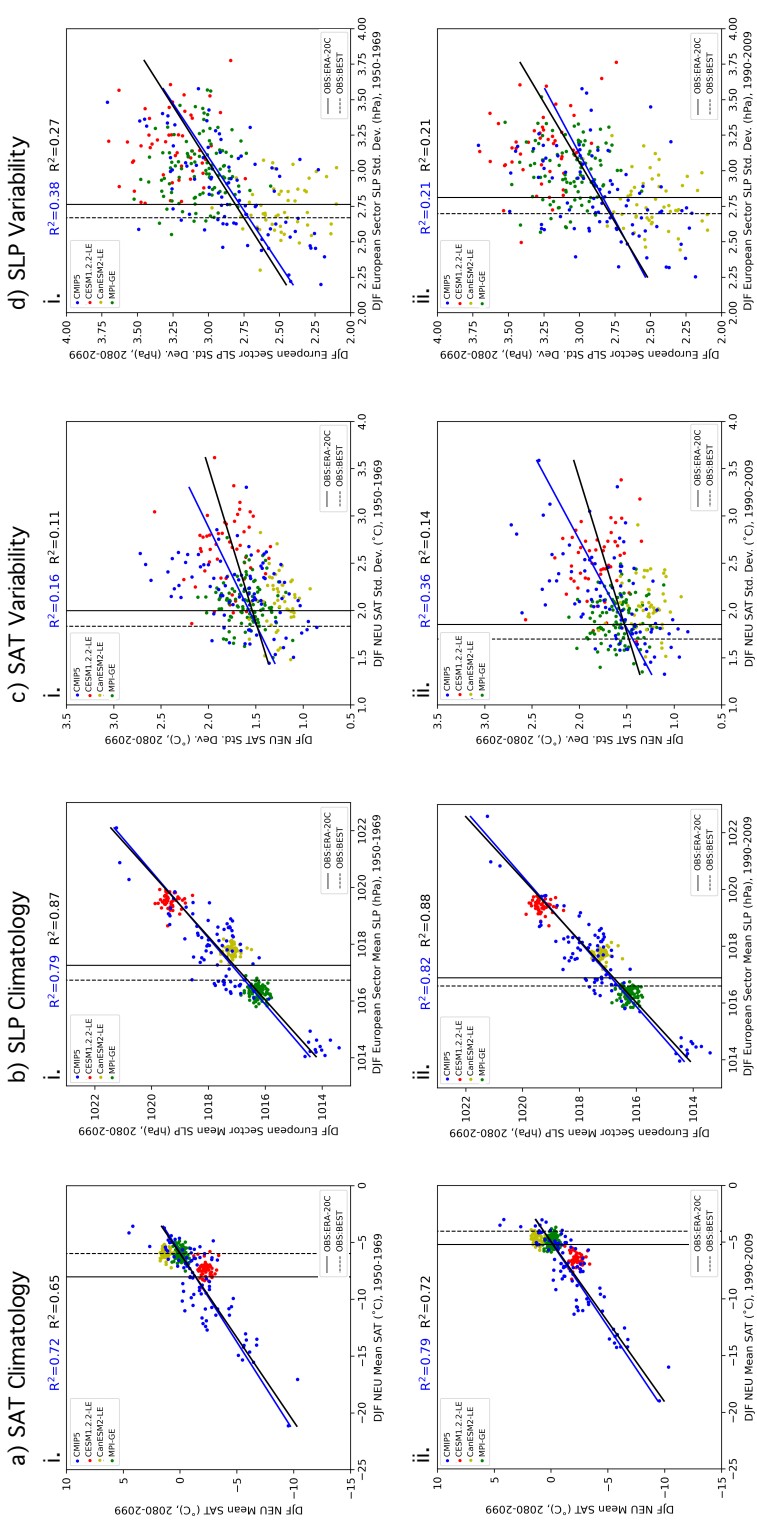

**Figure C1.** Predictor relationships in DJF comparing domain-averaged climate in two historical periods, (i) 1950-1969 and (ii) 1990-2009, to a future period, 2080-2099 in all panels. Observational estimates in the respective historical periods are indicated with a solid vertical line (ERA-20C SAT and SLP) and dashed vertical black line (BEST SAT and NOAA-20C SLP) in each panel. a) NEU SAT climatology ($^{\circ}$C), b) SLP climatology ($^{\circ}$C), averaged over the predictor region (hPa), c) NEU SAT standard deviation ($^{\circ}$C), and d) SLP standard deviation, averaged over the predictor region (hPa) are eight of the nine predictors used to determine RMSE member performance and independence in Figure 3. Least-squares regression fits (solid lines) and $R^2$ values computed for solely the CMIP5 output are shown in blue and computed for all output (CMIP5 and the three SMILEs) are shown in black.

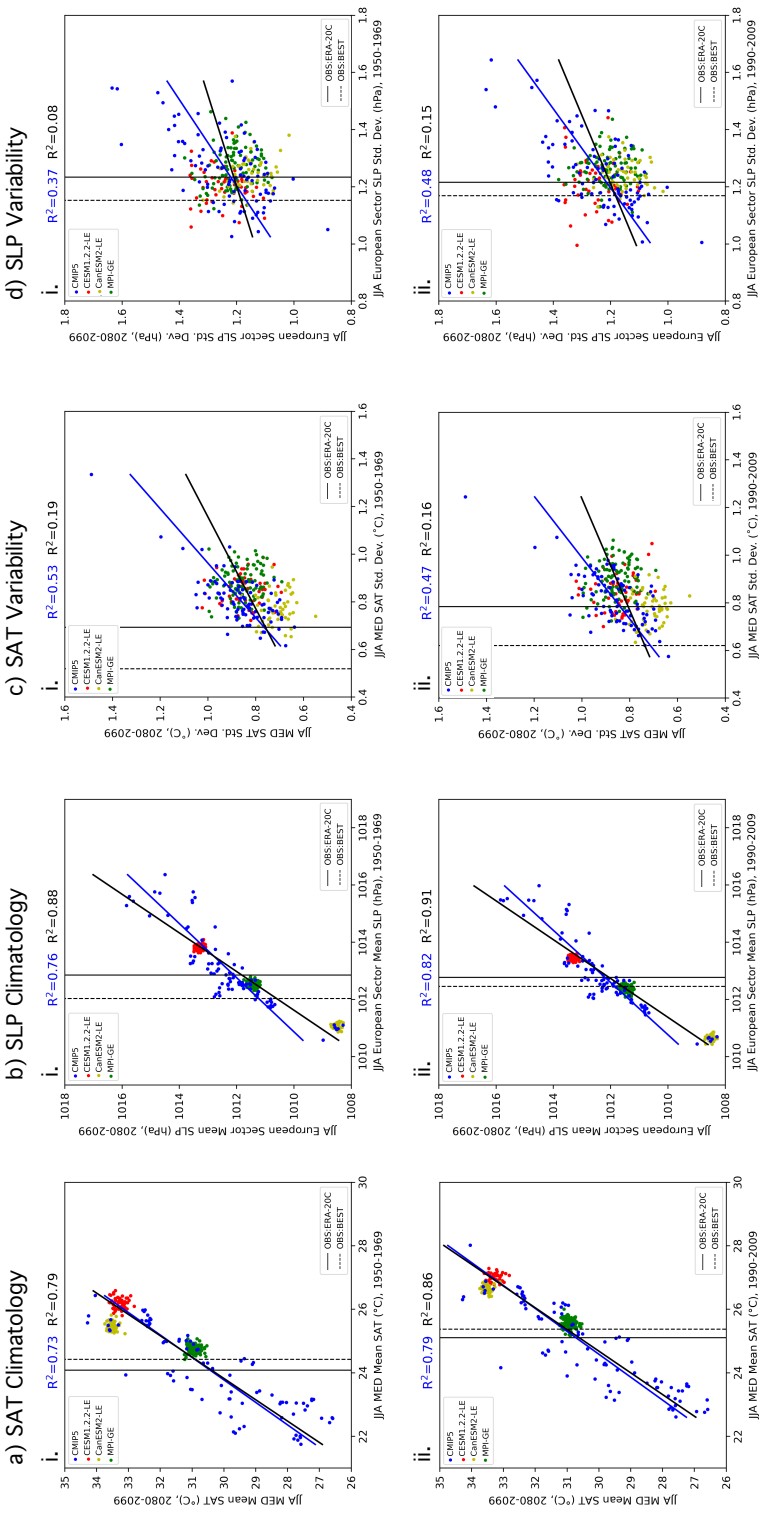

**Figure C2.** As in Figure C1, but for JJA and the MED region in columns a and c.

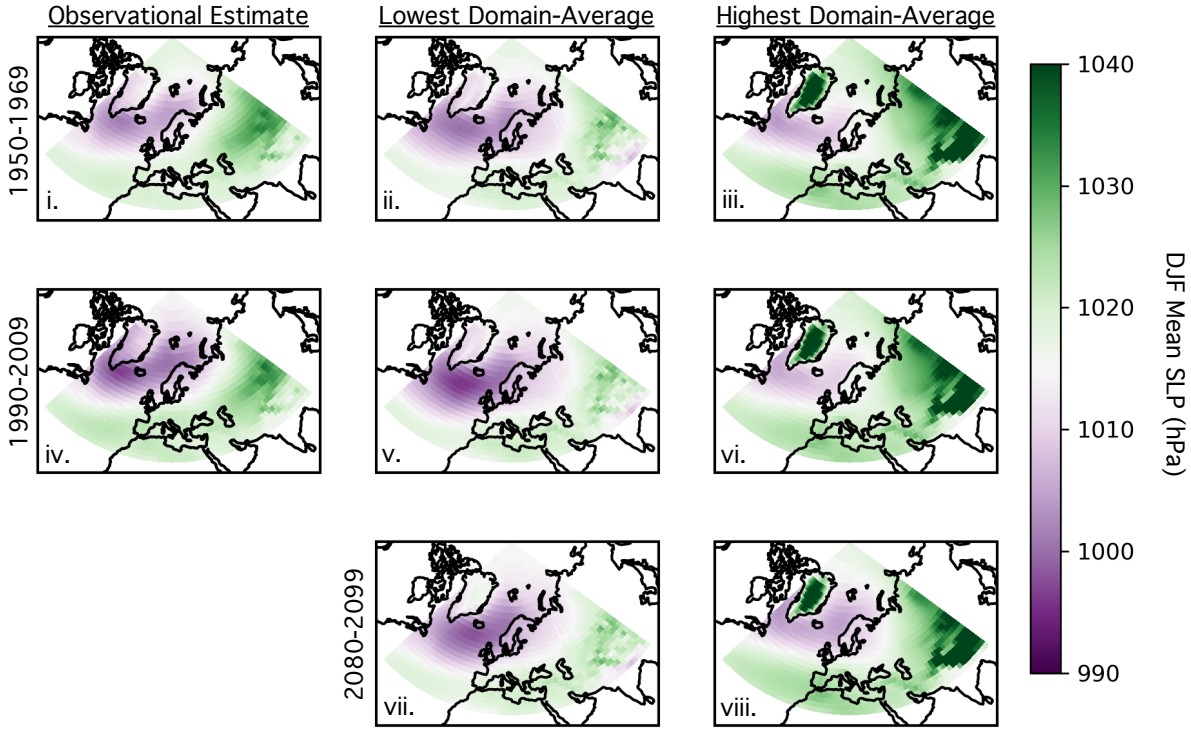

**Figure C3.** The spatial pattern of DJF SLP climatology for: 1950-1969 (i-iii), 1990-2009 (iv-vi), and 2080-2099 (vii-viii). The ERA-20C observational estimate of SLP climatology is shown in the left column (i,iv). The ensemble member with the lowest domain-average SLP climatology for the 1990-2009 historical period is shown in the center column (ii,v,vii). The ensemble member with the highest domain-average SLP climatology for the 1990-2009 period is shown in the right column (iii,vi,viii).

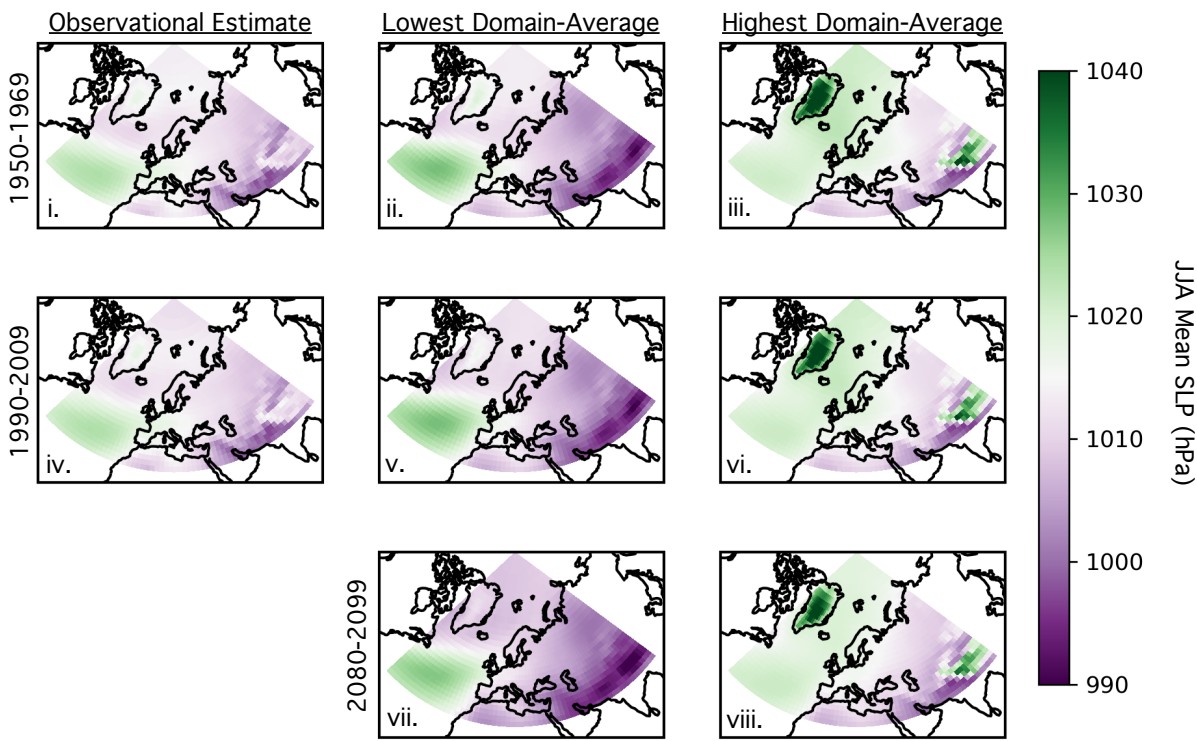

**Figure C4.** As in Figure C3, but for JJA SLP climatology.