# Peer review of "An investigation of weighting schemes suitable for incorporating large ensembles into multi-model ensembles"

_Earth System Dynamics, 2019_

## Referee Comment (RC1) · Anonymous Referee #1 · 10 Jan 2020

Review of "A weighting scheme to incorporate large ensembles in multi-model ensemble projections" by Merrifield et al.

In this paper, the authors describe the extension of a weighting scheme for multi-model climate projections described in previous works to incorporate single model initial condition large ensembles (SMILE). This weighting scheme uses a performance metric, based on the similarity of a simulation with observations and an independence metric, based on the similarity between simulations. Several properties of two variables (surface air temperature and sea level pressure) in the present climate are used to measure similarity. The authors intend to demonstrate the applicability and the usefulness of this weighting scheme with SMILEs, focusing on surface air temperature change over Northern Europe and the Mediterranean. They also discuss different properties

of the weights and some practical issues that may arise in such applications.

The subject of the paper is interesting and important, and there are some interesting analyses in this paper. It is well written and generally easy to follow. But I also think that the use of the proposed weighting scheme with SMILEs raises fundamental questions that are not addressed. As the incorporation of SMILEs in the weighting scheme is the novelty of the paper compared to previous works, these issues must be properly dealt with before the publication of the paper could be considered.

I am not sure that the authors can address these issues properly, as they are really intrinsic to the chosen approach, but I want to give them the opportunity to prove me wrong. I therefore recommend major revisions to the paper, but I may still recommend rejection of the paper at the next round.

Major comments

The notion of "independence" is perfectly defined in statistics and probability theory, but it is very ill defined when applied to climate models (which is not really acknowledged and discussed by the authors). In this paper, as in previous works, two models are considered more or less independent depending on the similarity of their results. Two models are considered "weakly" independent if their results are very similar and "strongly" independent if their results are very different. This is a hypothesis, and it should be discussed. The results of two "independent models" cannot be similar? Two independent models cannot converge towards the truth (if the models are close to the truth they will also be close from each other)? Overall, to my opinion, this hypothesis can make sense when dealing with multiple different models, and in any case there is no perfect theoretical and practical way to characterize model independence.

But I'm really bothered with this approach when dealing with members from the same model (only differing by initial conditions). I think that the attempt to use this weighting scheme with SMILEs illustrates some difficulties of the definition of independence in terms of similarity.

The members of a SMILE are independent in the statistical sense of the term, the only sense of independence that is well defined. But they are not independent in the approach proposed by the authors, and they can be more or less "independent" according the similarity of their results. For me, it is very problematic. If you roll a dice two times, you don't decide that two outcomes are "more independent" if you get a 4 and a 6 than if you get two 3. But it is basically what is done in the proposed method with SMILEs.

As an illustration of this issue: Imagine the particular case where we only have a single SMILE, and that we are interested by the distribution. Using the weighting scheme described in this paper is not correct in this case, right? We know that the SMILES members are independent and that each member should receive the same weight. It is what is done is all the studies based on a single SMILE. But the weighting scheme described in the paper would give different weights to different members. I think that the weighting scheme proposed by the authors (any weighting scheme) should hold seamlessly in a particular case like this one.

-Giving different performance weights to different members of the same climate model is also problematic, at a fundamental level, I think. The skill is intrinsic to the model, and not specific to a member of the model (once the memory due to initial conditions has disappeared). Whether a particular member of a SMILE is closer to the observations than another is purely accidental and says absolutely nothing on the realism of this particular member in the future climate.

-The baseline approach to which the weighting scheme is compared in this paper consists in giving an equal weight to all the members of the multi-member, multi-model ensemble (independently of the existence of other members of the same model in the ensemble). Obviously, it is a very bad approach, and nobody would do that, I think.

If we consider the models as independent and equally skilful, SMILE members can be easily incorporated in a multi-model ensemble, as it has been done for years, by

giving a weight to each member of a given model inversely proportional to the number of members of this model in the full ensemble. This approach is perfectly justified from a statistical standpoint (within the hypotheses made). (i) I think that the authors should use this approach as a baseline, to which they can compare their weighting scheme, and show the results obtained with this approach for example in Figure 3. (ii) Logically, the weights of an appropriate weighting scheme should tend towards the ones described above when the "hypothesis" of inter-model dependence and unequal realism is relaxed, I think. It is not the case with the weighting scheme described in the paper.

-I disagree with the interpretation of the results of dynamical adjustment in the paper. It is not possible to extract the "forced trend", even the "estimated forced trend" or the "radiatively-forced trend" with dynamical adjustment. Dynamical adjustment only allows separating the part of the trend that is due to large-scale atmospheric circulation from the part of the trend that is not due to large-scale atmospheric circulation. The "part of the trend that is due to atmospheric circulation" is not a correct estimation of the impact of internal variability, except in some particular cases. The variations in atmospheric circulation indeed can be forced, they are not necessarily of internal origin. There are quite a few papers on the detection and attribution of anthropogenic influences on large-scale atmospheric circulation, and there is a clear forced component (in the real sense of the term) in future circulation changes in many models. For this reason, the "part of the trend that is not due to large-scale atmospheric circulation" should not be named "forced trend", even "estimated forced trend". Additionally, it can bear the imprint of internal oceanic dynamics.

It is mainly a vocabulary issue here, as the interpretation of the results of dynamical adjustment does no really matter for the results discussed in the paper. Still, it is important to be correct.

Minor comments

l37. Parameterized processes are not the only reason for model uncertainty, I think. The dynamical cores can also be important in that context.

l53. As said in the major comments, dynamical adjustment cannot be used to quantify the impact of internal variability on climate variables. It can only be used to estimate the part of variability that is not driven by large-scale atmospheric circulation. It is completely different.

l55-56. You mean single "model" initial condition large ensemble and not single "member", right?

l85, data section I think it would be more logical to introduce the climate simulations (e.g. Table 1 etc.) before describing their result (Figure 1 etc.)

l96. ERA20C should not be used as observational reference for temperature. Only SLP and winds are assimilated in ERA20C, which leads to a sub-optimal representation of temperature variability. Not surprisingly, issues in regional temperature trends and low frequency variation exist in ERA20C. There are much better datasets to use for temperature. There is no need to use SLP and TAS from the same dataset to "assure consistency". Use the best dataset for each variable: normally, good observations from different sources are consistent. I also think that multiple observational datasets should be used in order to assess the impact of observational uncertainties.

l144. "that adds independent information". What is meant exactly by "independent information" (or "new" information, at some places)? It should be discussed, from a theoretical point of view.

l176. What "fit for purpose" means obviously depends on the purpose. I think it would be useful to state the purpose very precisely at this point (even if it can be inferred from other parts of the paper).

l185. I don't really understand how the RMSE distances are computed. You say that they are computed at each point before area averaging. RMSEs are not computed over

space but time? How do you compute the RMSE for a climatology at a given point? Please give the equations, it will be clearer.

l192-194. It is a reasonable idea when you consider two different models, but not when you consider two members of the same models. And in this paper two members of the same model are dealt with in the same way as two different models.

l200. There is no i (and ii) in Figure 2a. Please add the complete numbering of the sub-figures.

l207-213. The fact that the "estimated" forced trends are so different between members of the same model clearly shows that one should not talk of forced trends for the results of dynamical adjustment, preceded or not by "estimated". But I agree that independently of its name, it can be an interesting performance metric.

l214. "Internal variability": no, not necessarily (see major comments).

l228. Can you clarify what is meant by ""fair""?

l235 and Figure 3. I'm missing something: I don't understand how the weighted distributions (box-and-whiskers plots) are obtained, based on the weighting scheme described in the paper. It is not directly straightforward I think. Is it a parametric distribution, using the weighted variances and means and a Gaussian hypothesis? It does not seem to be the case as the whiskers are not symmetrical. Please explain how the percentiles are computed when using the weighting scheme.

l245. It would be interesting to add the results of the "classical" weighting scheme generally used when mixing SMILEs and multiple models (see major comments), that makes the hypothesis that the models are independent and equally skilful. It is a much better starting point for the comparison. Nobody in his right mind would add 200 members of the same climate model to the CMIP5 ensemble and compute the distribution without some basic weighting, right?

l254-255. This is rather obvious: see the previous comment.

l281-282. What criterion do you use to judge that the weighting is suitable? What is a suitable weighting scheme? It should be better discussed.

l285-320. I don't think that this analysis is that interesting. More important (and interesting) analyses are in Appendix.

l456-457. I don't see a test of the sensitivity to "sigma s" in Figure B2. You mean Figure B3? Should you not describe Figure B2 first?

---

## Referee Comment (RC2) · Anonymous Referee #2 · 14 Jan 2020

This paper explores the usefulness of an established model weighting procedure (following Knutti et al. 2017, Sanderson et al. 2017) for incorporating large ensembles of single-model projections into multi-model projections. The model weighting method is shown to produce reasonable results three large ensembles ("SMILEs") are combined with an ensemble of CMIP5 model runs. The paper is generally well written and the results interesting. Some improvements explaining the methods would be useful, but overall I think pending minor revisions the paper should be suitable for publication.

Main comments:

1. The selection of predictors is not completely convincing. I appreciate that the main purpose of the paper is to demonstrate that the weighting method is plausible for the type of ensemble considered, not to explore all possible choices of predictors. Nev-

ertheless Appendix C shows that variability (SLP and SAT standard deviation) shows a weak past/future relation, and Fig 2a suggests weak or no past/future relation for DJF NEU SAT estimated forced trend. Are the results sensitive to exclusion of these predictors?

2. Terminology of "independence weight": it's confusing that both numerator and denominator of equation (1) are called "weights". A more intuitive use would be that a "weight" is a quantity that's larger when the model run is given more weight, i.e. a stronger influence on the results. The term "weight" is used this way for the overall weight (left hand side of eq. 1) and the "performance weight", but not for the "independence weight". Could a different name be used or if not then could a note on this terminology at least be made clearly in Sec 3 (around l.155-165)?

3. The explanation of dynamical adjustment in Appendix A could be clearer. The meaning of Ns, Na and Nr isn't clearly explained. The description refers to the "observational record" but the method is also applied to models, both past and future. Is this appendix meant to be a standalone description of the method or is it assumed the reader is already familiar with the references? I would suggest to improve this description for benefit of completeness and also so that the reader isn't obliged to go to the references to have a basic understanding of the method. It could at least be described how the weights (beta_i) are determined.

Comments & suggestions by line number:

18: "increases linearly": maybe say "changes linearly". It seems unintuitive (at least to me) to describe the weight as increasing when it's actually the reciprocal of the "independence weight" that gets multiplied by the performance weight. This makes sense after reading Sec. 3, but someone reading just the abstract could find this confusing.

20: "subsetted ensemble" –> "subsetted ensemble of one model run per model"

45: "more-than-representative uncertainty" - what does this mean? Please clarify

and/or give a reference for this concept.

62: "ensemble, first," –> "ensemble. First,"

97-98: Don't most reanalyses provide both SLP and SAT?

100: ERA-20C doesn't assimilate surface temperature. Do you know that it's suitable for evaluating SAT trends? This could be tested by comparison with an observational dataset (HadCRUT4?).

109: "representative distribution" - what is this? The distribution is well defined for each model by virtue of the ensemble size. But is this term meant to suggest it's "representative" of the true variability? If not (and I'm not sure how that would be known), suggest change "representative" –> "well defined".

113: put quotes around "macro" (similar to "micro" at l.117)

121: "preindustrial" misspelled

122: "conditions, " –> "conditions: "

141: "and model, " –> "and model; "

167: "definition of climate, " –> "definition of climate: "

170: "fit for purpose" –> "fitness for purpose"

178: "trend" –> "estimated forced trend" (and perhaps add that meaning of this will be explained below)

191: "idea" –> "assumption"

205-206: Perhaps clarify here that the SMILEs reinforce the relationship in the sense that model-mean values (3 data points, one for each SMILE) support the relationship. It's not because the relationship is evident within the SMILEs, which it should not be since the relationship is due to model differences.

[Figure]

222: "trends, " –> "trends: "

223-225: The positive relation for the SMILEs is only for 3 models, and the CMIP5 relation is very weak. Overall this suggests no relation (across models) between past and future estimate forced trend for DJF NEU.

226: Not it's "bolstered" - perhaps more accurate to say that it's "robust to"? The relationship looks essentially the same for both cases in both panels of Fig 2b.

229: "use is" –> "use them"

235: "SMILE" –> "SMILEs"

235: Are these distributions over gridpoints? That is, at each gridpoint in the domain of interest, a weighted or unweighted mean over models is computed, and this contributes one member of the distributions shown in the Fig 2 box-whisker plots. Please clarify. If not then I'm not sure what the "weighted distribution" is.

253: "tail broadly" –> "tail is broadly"

288: "function number" –> "function of the number"

300: "a weight" –> "an overall weight"

324-325: Seems an odd way to say this. The distance-based independence measure used in the weighting is a proxy for model structural differences. Consider rephrasing as something like: "models have some independence from one another while members of a SMILE have none (in the sense of model structural uncertainty)".

367-368: Perhaps qualify this by saying it's a modest narrowing (according to Fig 3ai).

371-372: Again, this is a modest shift. Fig 3ai shows the 95th percentile of weighted ALL is only slightly higher than for unweighted ALL.

406: "target month" - meaning the target year, for the month under consideration?

457: I think you mean Figure B3.

Figure B3 caption: perhaps note that the unweighted distributions are the same in every panel, being shown for reference.

488: "domain-averages" –> "domain-averaged"

489: The "emergent constraint" here is that the model's climatological bias is more or less unchanged from past to future. Perhaps useful to also describe it in this simpler way?

---

## Author Comment (AC1) · 29 Feb 2020

**Response to "Review of "A weighting scheme to incorporate large ensembles in multi-model ensemble projections" by Merrifield et al."**

In this paper, the authors describe the extension of a weighting scheme for multi-model climate projections described in previous works to incorporate single model initial condition large ensembles (SMILE). This weighting scheme uses a performance metric, based on the similarity of a simulation with observations and an independence metric, based on the similarity between simulations. Several properties of two variables (surface air temperature and sea level pressure) in the present climate are used to measure similarity. The authors intend to demonstrate the applicability and the usefulness of this weighting scheme with SMILEs, focusing on surface air temperature change over Northern Europe and the Mediterranean. They also discuss different properties of the weights and some practical issues that may arise in such applications.

The subject of the paper is interesting and important, and there are some interesting analyses in this paper. It is well written and generally easy to follow. But I also think that the use of the proposed weighting scheme with SMILEs raises fundamental questions that are not addressed. As the incorporation of SMILEs in the weighting scheme is the novelty of the paper compared to previous works, these issues must be properly dealt with before the publication of the paper could be considered.

I am not sure that the authors can address these issues properly, as they are really intrinsic to the chosen approach, but I want to give them the opportunity to prove me wrong. I therefore recommend major revisions to the paper, but I may still recommend rejection of the paper at the next round.

Thank you for your comprehensive and thoughtful review of our paper; We really appreciate you taking the time to interrogate the underlying aspects of the weighting method, as your review has generated a lot of interesting discussion and new ideas for a path forward. We hope to be able to assuage some of your concerns in this response and in the subsequent revision of the manuscript.

**Major comments**

The notion of "independence" is perfectly defined in statistics and probability theory, but it is very ill defined when applied to climate models (which is not really acknowledged and discussed by the authors). In this paper, as in previous works, two models are considered more or less independent depending on the similarity of their results. Two models are considered "weakly" independent if their results are very similar and "strongly" independent if their results are very different. This is a hypothesis, and it should be discussed.

We agree that we should include a more comprehensive discussion about our hypothesis of independence as a measure of whether a model or simulation provides additional information by having a distinguishable representation of historical climate (which is indeed not the formal statistical definition of independence). The reasoning behind defining independence as the RMSE distance between models, based on a collection of historical features, is to identify and reduce the influence of shared model biases on the uncertainty distribution of interest. An example is temperature mean state: models that share a warm bias in the Mediterannean may have dried into a land-atmosphere feedback regime different from what has been observed, which then amplifies warming. Additionally, the choice to use RMSE distance allows a measure of independence that doesn't rely on *a priori* knowledge of code sharing, branched development, etc., which, to some extent, renders models with different names not independent as well.

Following this suggestion, we have added a discussion about independence assumptions to the paper to accompany the new analysis on how different independence assumptions affect the weighting (new versions of Figure 3 and 4).

Figure 3. (a) Box-and-whisker plot showing how the five weighting strategies effect the distributions of DJF NEU SAT change ( $\Delta$ , [2080-2099]-[1990-2009]) for the CMIP5 ensemble (blue) and ALL ensemble (CMIP5 with the 3 SMILEs; gray). The box element spans the 25th to 75th percentile of the distribution; mean SAT change is indicated by the horizontal line within the box. The whisker element spans the 5th to 95th percentile. b) As in a), but for JJA MED SAT change. c) The contribution of SMILE and CMIP5 members to the DJF NEU ALL ensemble under different weighting strategies, in terms of fraction of total weight. d) As in c), but for the JJA MED ALL ensemble.

---

## Author Comment (AC2) · 29 Feb 2020

**Response to "Review of "A weighting scheme to incorporate large ensembles in multi-model ensemble projections" by Merrifield et al.""**

This paper explores the usefulness of an established model weighting procedure (following Knutti et al. 2017, Sanderson et al. 2017) for incorporating large ensembles of single-model projections into multi-model projections. The model weighting method is shown to produce reasonable results three large ensembles ("SMILEs") are combined with an ensemble of CMIP5 model runs. The paper is generally well written and the results interesting. Some improvements explaining the methods would be useful, but overall I think pending minor revisions the paper should be suitable for publication.

```
Thank you for taking the time to review our manuscript, we really
appreciate your feedback and are happy to hear that you are
interested in some of the results. We hope you similarly find our new
analysis associated with alternative independence assumptions
interesting and that we are able to address in the revised version of
the manuscript.
```

**Main comments:**

1. The selection of predictors is not completely convincing. I appreciate that the main purpose of the paper is to demonstrate that the weighting method is plausible for the type of ensemble considered, not to explore all possible choices of predictors. Nevertheless Appendix C shows that variability (SLP and SAT standard deviation) shows a weak past/future relation, and Fig 2a suggests weak or no past/future relation for DJF NEU SAT estimated forced trend. Are the results sensitive to exclusion of these predictors?

```
It is a good point that not all predictors we have chosen have strong
emergent relationships, and we will add more discussion about
predictor selection, both in the main text and in the appendix. We've
revised the following in the main text:
```

```
"The assumption is that if a model accurately represents an aspect of
historical climate, it is likely to realistically represent relevant
physical processes and therefore is likely to provide a reliable
future projection. If a model is significantly biased with respect to
observed climate, its future representation of climate may be cause
for concern (Knutti et al. 2017). For these tendencies to hold, a
statistical relationship between the historical and future climate
feature of interest must exist. In the absence of a strong
relationship, predictors serve to add degrees of difference between
members which helps to ward against overconfident weighting."
```

The tasSTD predictor is considered to be one of the better predictors in terms of correlation with the end-of-century warming target (as in Lorenz et al. 2018) over both periods for the DJF NEU. For the JJA MED case, several members of CMIP5 have more-than-observed SAT and SLP variability, which features we wanted to be reflected in the performance weight as biases can indicate issues with physical processes (i.e., land-atmosphere interactions). Ultimately, though, if a predictor does not have a strong emergent relationship, its inclusion simply adds a bit of noise into the distances but doesn't strongly affect the CMIP5 weighting (below).

[Figure]

For the ALL ensemble, using only climatological predictors further narrows the distribution because the SMILEs tend to have near observed climatology and therefore a higher performance weight than other CMIP5 members. This illustrates why it tends not to be a good practice to use one or few predictors for performance with this method. We try to avoid situations of overconfidence in future results, as no single emergent relationship is really indicative of a model's performance.

However, climatology has proven to be a better indicator of dependence. We've added analysis of this in new versions of Figure 4 and 5.

[Figure]

**Figure 4.** (a-di) The RMSE independence scaling of the SMILEs and CMIP5 ensemble members, shown in the order listed in Table 1. Panels ai and ci show the scaling computed from the 9 predictors used in the original DJF and JJA RMSE distance weighting respectively. Panels bi and di show the scaling computed from 2 predictors: global land SAT and European sector SLP climatology over the 1950-2010 period for DJF and JJA respectively. (a-d ii) The sorted RMSE distance between member 1 of the CESM1.2.2-LE and all other members of the ALL ensemble. (a-d iii) As in ii, but for CanESM2-LE member 1. (a-d iv) As in ii and iii, but for MPI-GE member 1.

[Figure]

**Figure 5.** a) Scatter plot showing how ALL ensemble members distribute in the DJF European sector SLP climatology / DJF Global Land SAT climatology predictor space. Select clusters within the CMIP5 ensemble (blue) are labelled by model name. The CESM1.2.2-LE is indicated in red, the CanESM2-LE is indicated in yellow, and the MPI-GE is indicated in green, consistent with other figures. b) As in a), but for the JJA European sector SLP climatology / JJA Global Land SAT climatology predictor space.

2. Terminology of "independence weight": it's confusing that both numerator and denominator of equation (1) are called "weights". A more intuitive use would be that a "weight" is a quantity that's larger when the model run is given more weight, i.e. a stronger influence on the results. The term "weight" is used this way for the overall weight (left hand side of eq. 1) and the "performance weight", but not for the "independence weight". Could a different name be used or if not then could a note on this terminology at least be made clearly in Sec 3 (around l.155-165)?

Thank you for pointing this out, this distinction of using "weight" to refer to the directly proportional performance term and "scaling" to  refer to the inversely proportional independence term will definitely help with the clarity and will be henceforth used.

3. The explanation of dynamical adjustment in Appendix A could be clearer. The meaning of Ns, Na and Nr isn't clearly explained. The description refers to the "observational record" but the method is also applied to models, both past and future. Is this appendix meant to be a standalone description of the method or is it assumed the reader is already familiar with the references? I would suggest to improve this description for benefit of completeness and also so that the reader isn't obliged to go to the references to have a basic understanding of the method. It could at least be described how the weights (beta_i) are determined.

We apologize for leaning too heavily on prior works for the basic understanding of the dynamical adjustment methodology. We have revised the section the as follows, an we hope you find the description clearer:

"To obtain estimated forced trends in SAT, a method of dynamical adjustment, based on constructed circulation analogues, is used (Deser et al., 2016; Lehner et al., 2017; Merrifield et al., 2017; Guo et al., 2019). Dynamical adjustment provides an empirically-derived estimate of the SAT trends induced by atmospheric circulation variability; removal of this circulation-driven component from a SAT record thus reveals an estimate of the SAT trend associated with thermodynamic processes and radiative effects. Dynamical adjustment relies on the ability to reconstruct a monthly mean circulation field, which we represent with sea level pressure (SLP) as in Deser et al. (2016), from a large set of analogues. Here, SLP analogues are selected from 60 possible choices (from the period 1950-2010), excluding the target month, and the method is therefore referred to as the "leave-one-out" method of dynamical adjustment.

SLP fields in SMILE members, CMIP5 ensemble members, and the observational estimates ERA-20C and NOAA-20C are constructed in this manner for target months in the 1950-2010 period. For model years 2011-2099, analogues are selected from the entire 1950-2010 period. No notable trends in SLP have been identified over this period in previous dynamical adjustment studies (Deser et al., 2012, 2016; Lehner et al., 2017).

It is important to acknowledge that because of the paucity of analogue choices in leave-one-out dynamical adjustment, the term "analogue" is a bit of a misnomer. The term evokes the idea of a match, though in practice, analogues may not closely resemble the target. For convenience, we will continue to refer to the months used in target SLP construction as "analogues", but we do so with the understanding that target and analogue patterns may differ over the selection domain. A month is determined to be an analogue of the target month if the Euclidean distance between target and analogue SLP is small. Euclidean distance is computed at each grid point and averaged over the European sector domain also used for SLP predictors (25-90°N, 60°W-100°E). This selection metric, therefore, does not require an analogue to match the target month spatially over the whole domain. This is necessary because, with 60 possible options, it is statistically unlikely that a "perfect" analogue will exist for a particular target month. van den Dool (1994) found that it would take on the order of 10 years to find two Northern hemisphere circulation patterns that match within observational uncertainty. With this in mind, a smaller than hemispheric domain and an iterative averaging schemes are employed to make the most of "imperfect" analogues available (Wallace et al., 2012; Deser et al., 2014, 2016). Once the Euclidean distances are determined, 50 closest SLP analogues are chosen, and the iterative process of selecting 30 of 50 SLP analogues and optimally reconstructing a target SLP field $X_h$ commences. The optimal reconstruction of target SLP is mathematically equivalent to multivariate linear regression; each analogue is assigned a weight ($\beta$) such that a weighted linear combination of analogues produces a least-squares estimate of the target SLP. $\beta$ is computed through a singular value decomposition of a column vector matrix $X_c$ containing the 30 selected analogues and can also be estimated using through a Moore-Penrose pseudoinverse:

$$\beta = [(\mathbf{X}_c^T \mathbf{X}_c)^{-1} \mathbf{X}_c^T] \mathbf{X}_h \tag{A1}$$

The analogue weighting scheme ensures that analogues which are further from (closer to) the target, in a Euclidean distance sense, contribute less (more) to the constructed SLP field.

After the target SLP field is constructed, the **β** values derived for each SLP analogue are applied to their corresponding monthly-averaged SAT fields. Prior to the application of weights, a quadratic trend representing anthropogenic warming is removed from the SAT record at each point in space. The purpose of this detrending is so that months picked from the end of the record do not contribute higher SAT anomalies simply because of the anthropogenically forced warmer background climate, even if the SLP patterns are the same (Lehner et al. 2017). Detrending strategies are further discussed in Deser et al. (2016). The weighted, detrended SAT fields are then used to construct a dynamic SAT anomaly field for the target month. SLP, which is a representative of low-level atmospheric circulation, and SAT are physically related; SLP-derived weights are applied to SAT to empirically construct that relationship. Conceptually, dynamic SAT anomalies are those that would occur given the attendant circulation pattern. The second through fifth steps of dynamical adjustment (selection of 30 of 50 SLP analogues, optimal reconstruction of target SLP, and construction of dynamic SAT) are then repeated 100 times, following Lehner et al. (2017). The dynamic component of SAT in the target month is the average of the 100 constructions. It is then subtracted from SAT in the target month to find the residual thermodynamic component of SAT, used as an estimate of the regional SAT response to surface processes and radiative forcing. The trend of the residual thermodynamic SAT component is used as a predictor in this study; trend is computed at each land grid point in the predictor domain and subsequently area-averaged."

**Comments & suggestions by line number:**

18: "increases linearly": maybe say "changes linearly". It seems unintuitive (at least to me) to describe the weight as increasing when it's actually the reciprocal of the "independence weight" that gets multiplied by the performance weight. This makes sense after reading Sec. 3, but someone reading just the abstract could find this confusing.

Thank you, corrected.

20: "subsetted ensemble" –> "subsetted ensemble of one model run per model"

Corrected.

45: "more-than-representative uncertainty" - what does this mean? Please clarify and/or give a reference for this concept.

We have revised this sentence to:

"Known biases associated with cloud processes, land-atmosphere interactions, and sea surface temperature (e.g. Boberg and Christensen, 2012; Li and Xie, 2012; Pithan et al., 2014; Merrifield and Xie, 2016) may result in more uncertainty in projections of future climate than is warranted given our understanding of the climate system (Vogel et al. 2018). Using expert judgement to weight or select multi-model ensemble members based on process- or region-specific metrics of performance has been shown to justifiably constrain uncertainty (e.g. Abramowitz et al., 2008; Knutti et al., 2017; Lorenz et al., 2018)."

62: "ensemble, first," –> "ensemble. First,"

Corrected.

97-98: Don't most reanalyses provide both SLP and SAT?  100: ERA-20C doesn't assimilate surface temperature. Do you know that it's suitable for evaluating SAT trends? This could be tested by comparison with an observational dataset (HadCRUT4?).

We have added an additional dataset to better assess observational uncertainty: the Berkeley Earth Surface Temperature (BEST) product and NOAA-20C SLP reanalysis V3. We were unfortunately not able to use HadCRUT4 without decimating all other fields onto its 5˚x5˚ grid. But we hope that the observational datasets we did select serve to establish observational uncertainty/suitability.

109: "representative distribution" - what is this? The distribution is well defined for each model by virtue of the ensemble size. But is this term meant to suggest it's "representative" of the true variability? If not (and I'm not sure how that would be known), suggest change "representative" –> "well defined".

We agree that "well-defined" is a much better choice for describing SMILEs distributions. We have changed the data section quite a bit, but have used this recommendation as follows:

"The multi-model CMIP5 ensemble (Fig.1 blue) has more spread than the single model SMILEs, demonstrating that model uncertainty does rise above well-defined estimates of internal variability in the two European regions and seasons considered."

113: put quotes around "macro" (similar to "micro" at l.117)

Punctuation updated, thank you.

121: "preindustrial" misspelled

Thank you, Corrected.

122: "conditions, " –> "conditions: "

Punctuation updated, thank you.

141: "and model, " –> "and model; "

Thank you for the catch. This paragraph has been heavily revised in the revision and the relevant sentence has been removed.

167: "definition of climate, " –> "definition of climate: "

This sentence has been split up in the revision and now reads:

"Both the performance weight used in weighting strategies 2-5 and the independence scaling used in strategy 5 are based on a chosen definition of climate. A model's performance is based on its ability to reproduce observed climate and a member's independence is based on how much its climate differs from the climate in other members."

170: "fit for purpose" –> "fitness for purpose"

Changed.

178: "trend" –> "estimated forced trend" (and perhaps add that meaning of this will be explained below)

Thank you for the catch. This has been changed as follows:

"...and a 50-year derived SAT trend (estimated residual thermodynamic
trend; described in more detail in subsequent paragraphs) for the
period of 1960-2009"

191: "idea" –> "assumption"

Changed.

205-206: Perhaps clarify here that the SMILES reinforce the relationship in the sense that
model-mean values (3 data points, one for each SMILE) support the relationship. It's not
because the relationship is evident within the SMILEs, which it should not be since the
relationship is due to model differences.

Thank you, this sentence has been revised as follows:

"In contrast, a relationship emerges in summer, a season with less
midlatitude SAT variability, between 1960-2009 and 2050-2099 European
SAT trends. The linear relationship is reinforced by the SMILEs in a
model mean sense, i.e., the three new models added to the CMIP5
ensemble support the relationship (Fig.2bi). It is not evident within
the SMILEs themselves, which reflects that the relationship is due to
model differences not the behavior of individual members."

222: "trends, " –> "trends: "

Changed.

223-225: The positive relation for the SMILEs is only for 3 models, and the CMIP5 relation is
very weak. Overall this suggests no relation (across models) between past and future estimate
forced trend for DJF NEU.

You are correct, we've revised the sentence to read:

"The addition of the SMILEs then introduces a slightly positive
relationship between past and future responses (Fig. 2aii, black
line) not apparent in the CMIP5 ensemble (Fig. 2aii, blue line),
though no strong relationship emerges from variability in either
case."

226: Not it's "bolstered" - perhaps more accurate to say that it's "robust to"? The relationship
looks essentially the same for both cases in both panels of Fig 2b.

Thank you, changed.

229: "use is" –> "use them"

Thank you, corrected.

235: "SMILE" –> "SMILEs"

Corrected.

235: Are these distributions over gridpoints? That is, at each gridpoint in the domain of interest, a weighted or unweighted mean over models is computed, and this contributes one member of the distributions shown in the Fig 2 box-whisker plots. Please clarify. If not then I'm not sure what the "weighted distribution" is.

It's a good point that it wasn't entirely clear on what scale the weights were computed on. To address this, we've added the following to the weighting section:

"To compute the aggregate distance metrics from 9 predictors, all predictor and observational fields are bilinearly interpolated to a shared 2.5° x 2.5° latitude-longitude grid. The predictors are then time-aggregated, with the mean or standard deviation computed over the periods 1950-1969 and 1990-2009, and the estimated residual thermodynamic trend computed over the period 1960-2009. For each time-aggregated predictor, the differences between the observed mean value and member value (or member value and member value) are computed at each grid point and subsequently squared. The squared differences are then area-averaged over the predictor domain and square-rooted to obtain an RMSE distance for observed-member and member-member pairs. For each predictor, the resulting distributions of observed-member and member-member RMSEs are then normalized by their mid-range value ([maximum + minimum]/2), such that the distance for each of the nine predictors are on the same order of magnitude and can be combined into a single $D_i$ (Figure B1) and $S_{ij}$ (Figure B2) for each member."

And the following at the aforementioned location:

"Two ensembles are considered, one comprised solely of CMIP5 members (CMIP5; distribution of 88 values) and one comprised of all available members from CMIP5 and the three SMILEs (ALL; distribution of 288 values."
* * *
**From this point, we have made a complete overhaul of the Results section and the relevant sentences no longer exist. We will take all of the following recommendations forward in the revision, particularly to describe distributional shifts. Thank you for all the specific feedback!**

253: "tail broadly" –> "tail is broadly"
288: "function number" –> "function of the number"
300: "a weight" –> "an overall weight"
324-325: Seems an odd way to say this. The distance-based independence measure used in the weighting is a proxy for model structural differences. Consider rephrasing as something like: "models have some independence from one another while members of a SMILE have none (in the sense of model structural uncertainty)".
367-368: Perhaps qualify this by saying it's a modest narrowing (according to Fig 3ai).
371-372: Again, this is a modest shift. Fig 3ai shows the 95th percentile of weighted ALL is only slightly higher than for unweighted ALL.
406: "target month" - meaning the target year, for the month under consideration?
457: I think you mean Figure B3.
488: "domain-averages" –> "domain-averaged"
489: The "emergent constraint" here is that the model's climatological bias is more or less unchanged from past to future. Perhaps useful to also describe it in this simpler way?

Figure B3 caption: perhaps note that the unweighted distributions are the same in every panel, being shown for reference.

To make the figure less busy, we have removed the unweighted distributions from this figure.

---

## Author Response (AR1)

**Anna L. Merrifield**
*Universitätstrasse 16, 8092 Zürich*
*Switzerland*
✆ *+41 (78) 660 6670*
✉ *anna.merrifield@env.ethz.ch*

**Earth System Dynamics**                                                       May 13, 2020
*An EGU Journal*

Dear Dr. Milinski,

We have resubmitted the article *A weighting scheme to incorporate large ensembles in multi-model ensemble projections* by Anna L. Merrifield, Lukas Brunner, Ruth Lorenz, Iselin Medhaug, and Reto Knutti (please note additional author, Iselin Medhaug) to be considered for publication in Earth System Dynamics. We have made substantial changes to the document in line with reviewer comments and have provided a side-by-side document comparison to try to capture the scope of changes. Please let us know if an additional track change document is necessary and we will do our best to provide one.

We have followed through with our response to the reviewers (submitted prior) and hope you find your two fundamental questions addressed: (1) Similarity between the output from two simulations can either indicate that the underlying models are not independent and share biases, or indicate that two independent models are converging on the truth and (2) Realisations generated by the same model should have identical performance. For (1), we explore known vs. model output-assumed dependencies and propose a separate set of predictors that are independent of performance, such that models are not penalized for converging to truth. For (2), we fix initial condition member performance to be the same and treat initial condition ensembles as models throughout the manuscript. Thank you for your consideration and please let us know if an additional revised point-by-point response to the reviewers is necessary.

Sincerely,

**Anna L. Merrifield**

**Compare Results**

| Old File: | | New File: |
|---|---|---|
| **ESD_Merrifield_Old.pdf** | versus | **ESD_Merrifield_Rev.pdf** |
| **30 pages (16.20 MB)** | | **36 pages (14.36 MB)** |
| 13.05.20, 15:42:39 | | 13.05.20, 16:07:54 |

**Total Changes**

**437**

Text only comparison

**Content**

| 196 | Replacements |
|---|---|
| 135 | Insertions |
| 106 | Deletions |

**Styling and Annotations**

| 0 | Styling |
|---|---|
| 0 | Annotations |

Go to First Change (page 1)

[revised manuscript text omitted]

---

## Author Response (AR2)

**Response to: suggestions for revision or reasons for rejection (will be published if the paper is accepted for final publication)**

I really appreciate the responses to my comments and the major modifications made to the paper. I think the paper is more interesting now, with a very thorough investigation of issues associated with weighting, that goes beyond the simple incorporation of SMILES in ensemble projections (by the way, I'm not sure that the title is really optimal now). The paper will be suitable for publication after minor revisions.

We would like to thank you for your constructive and thought-provoking reviews both in this round and in the round before. They led to a lot of interesting discussions on our end and hopefully, a paper that will be of interest to a broader audience than before. We do plan to change the title to: An investigation of weighting schemes suitable for incorporating large ensembles into multi-model ensembles. Thank you for the recommendation to change the title as well!

The line numbers refer to the version with highlighted changes in the author response.

L4 "SMILES represent internal variability…" The formulation seems strange to me.

Thank you for pointing this out. We've changed the sentence to read:

L3-4: "SMILEs allow for the quantification of internal variability…"

L124 You could say a word on these differences of physics (they are quite limited actually)

Absolutely, we've added the following description:

L124-127: "Additionally, for the GISS-E2-H and GISS-E2-R experiments, NOAA GISS provides members from 3 physics-version ("p") setups that differ in atmospheric composition (AC) and aerosol indirect effects (AIE) (Miller et al., 2014). We treat the 3 setups as follows: p1 (prescribed AC and AIE) and p3 (prognostic AC and partial AIE) members are treated as 2 member IC ensembles and the p2 member (prognostic AC and AIE) is treated as a single member representation (Table 1)."

Figure 1. What is shown for CMIP5? Statistics on all the members, with equal weighting, without taking into account that some models provide several IC members? It should be noted in the

legend as it is not an approach that one would normally used, with maybe a reference to the discussion in section 3.1. Spread: please say in the legend how the spread is calculated in practice. There are many ways to define a spread.

This is a great point; thanks for bringing it up. In line with this recommendation, we've decided to update Figure 1 to show ensemble spread as the 5th-95th percentiles of each distribution (rather than making any sort of gaussian assumption). We've changed the caption of Figure 1 to read:

"Observational estimates (OBS; gray), the CMIP5 ensemble (blue), and the three SMILEs: CESM1.2.2-LE (red), CanESM2-LE (yellow), and MPI-GE (green) evaluated in this study, shown in terms of area- and seasonally-averaged absolute surface air temperature timeseries (SAT; ˚C). The two OBS datasets, ERA-20C Temperature and the Berkeley Earth Surface Temperature (BEST) product, are shown in solid gray and dashed gray respectively. Their average, used to determine member performance, is shown in solid black. For the CMIP5 and three SMILEs, the ensemble means across members are shown in solid color; the shading indicates the 5th-95th percentile of each distribution as a measure of ensemble spread. Note that the CMIP5 ensemble is a multi-model, multi-initial condition member ensemble of 88 members from 40 (named) model setups, not the "one model, one vote" ensemble often used in multi-model ensemble studies. Panel a shows projections for Northern European Winter (DJF NEU) and panel b shows projections for Mediterranean summer (JJA MED) SAT.  The number of members in each ensemble is indicated in parenthesis in the legend."

We've also changed the main text to read:

L152-153: "The CMIP5 ensemble and three SMILEs are shown in terms of their respective ensemble means and spreads (represented by the 5th-95th percentile of each distribution) in Figure 1,..."

L218. For wiIII. I don't understand the 1/N2 factor. It should be 1/N. Is this just a mistake in the written equation (the title of the subsection is 1/N scaling after all) or are the analyses impacted?

We agree that our choice to write the equation with a $1/N^2$ is unnecessarily confusing. The $1/N^2$ comes from the combination of the $1/N$ scaling and the $1/N$ in the average formula applied to the IC member performance weights (such that each IC member receives an

identical performance weight). We've changed the formula wiIII and
wiIV to more directly reflect the namesake 1/N scaling.

L236. Does each model get a unique performance weight as in section 3.3 or are different
performance weights given to each IC member of a model? It is not clear.

This is an important methodological point. We've revised the
paragraph to read:

L234-241: "Finally, the fifth weighting strategy operates under the
assumption that independence cannot necessarily be determined by
model name, but shared biases in simulating historical climate can
give an idea of dependence that comes from differently named models
sharing ideas and code. Instead of relying on knowledge of model
origin, the RMSE weighting (wiV) initially proposed by Knutti et al.
(2017) relies solely on model output to determine a model's overall
weight. It features an independence scaling based on RMSE distance
metrics in addition to the RMSE-derived performance weights.  For
results to be compatible with past assessments of this weighting
scheme (e.g. Lorenz et al., 2018; Brunner et al., 2019), we assign
each member their unique performance weight (as computed in wiII)
even if they are IC ensemble members. This puts the RMSE weighting in
contrast to the 1/N scaling approaches which ensure IC ensemble
members have identical weights."

Line 378-379. It is not totally obvious to me: if a model is almost the truth, is almost perfectly
realistic, and all the other models are wrong, why would it be problematic for this model to be
over-represented? How do we decide in practice that a model is over-represented or not?

You are right to point out that in order to make the claim a model is
over-represented, one would need to do something along the lines of
out-of-sample testing. We've rephrased the section to reflect that
the outsized contribution of the MPI-GE comes not from it being so
much higher performing than other models, but rather from there being
100 high performing members present:

L381-389: "Uncertainty in the DJF NEU ALL ensemble is constrained
both by the performance weight diminishing the contribution of CMIP5
members and because MPI is one of the highest performing models based
on the chosen DJF predictors. The high performing MPI-GE receives
65.8% of the total ALL ensemble weight, though individual MPI-GE
members only receive up to three times more weight than the averaged

assigned weight. The aggregate impact of 100 high performing members, however, is outsized and results in the narrowing of the performance weighted end-of-century warming distribution. The narrowing does not reflect the increased certainty that comes from the agreement of independent entities within the ensemble. Instead, it exemplifies that there is a need for an independence assumption in order to avoid the outsize influence that comes from being both historically realistic and numerously represented in the ensemble."

**Line 386-387. I'm not sure I understand that reasoning. The spread could be the same just coincidentally.**

It's true they could be the same coincidentally. We hoped to convey that the equally weighted distribution was too narrow because of the majority influence from just three models. Because the performance distribution was as narrow as the equally weighted distribution, we extrapolated that it was likely somewhat over-confident as well. The line of thinking is not crucial to the overall message of the paper, though, and so we've revised the section to read:

L390-394: "For JJA MED SAT change, the performance weight reduces the contribution of the three SMILEs to the ALL distribution in comparison to the equal weighting case, with the largest reduction made to CanESM2-LE contribution (17.4% to 7.4%; Fig.3d). However, the three SMILEs (three independent entities) still receive 51% of the total JJA MED ALL ensemble weight, their contributions again augmented by numerous representations. As in the equal weighting case, the JJA MED ALL performance-weighted ensemble mean is still modestly shifted towards more end-of-century warming than its JJA MED CMIP5 counterpart. This reflects the above CMIP5-average SAT change of the CESM1.2.2-LE and the CanESM2-LE in Mediterranean summer."

**L389. "In light of the clear necessity": Not that clear to me.**

Thanks for pointing out the need for clarification. We've changed the opening to:

L396: "In an effort to more appropriately handle the mix of models and IC members present in the ALL ensemble,..."

L567: "A weighting scheme, such as the one assessed here, is thus ideal for providing justifiable estimates of uncertainty…" Which weighting scheme? Several weighting schemes are discussed, and for me it is not obvious which one is the best among the last three. This paper actually shows that different weighting schemes that make sense can lead to different results in terms of spread and even response. It is therefore still very difficult to provide justifiable "estimates of uncertainty". And as discussed in Appendix B some ad-hoc choices have to be made: values of sigma d and s etc. I really appreciate the different tests used by the authors to define these parameters, they did a very good job, but some subjectivity remains.

You bring up a really important point that the "justifiable estimate of uncertainty" is still very much a matter of debate. Our closing didn't really reflect that and further, it didn't highlight the main finding (reconciling that the RMSE scaling in its first conception wasn't able to distinguish between IC members and models). In line with this, we've revised the last paragraph to read:

L574-579: "For more conventional multi-model ensembles that may include just a few initial condition ensemble members amongst the models, results may be less sensitive to choices underpinning the independence scaling. When large ensembles are included, however, it becomes clear that an independence scaling, such as the RMSE global predictor scaling presented here, that both scales known dependencies appropriately (i.e., 1/N for IC ensemble members) and assigns a degree of independence to remaining members is necessary. Such an independence scaling will be a useful tool with which to assess uncertainty in the combined multi-model, multi-initial condition ensemble member CMIP6 ensemble."

L625. I'm not sure to understand exactly how it is done. How the prediction intervals corresponding to each truth are calculated in practice?

Thank you for bringing this point up, we were much too brief in our description of the perfect model test. Because the performance weighting was not the primary consideration of this study, we compared a simplified perfect model test to the perfect model test used in previous studies. The two versions are similar (using each model as "truth", then predicting that "truth" using the other models) and are consistent in the σD they give. We hope you find the updated description (and example plot) of the simplified method to be more helpful:

L628-645: "Determining the shape parameters $\sigma D$ and $\sigma S$ is an important step in the RMSE weighting process (Knutti et al. 2017). $\sigma D$ can be set using a perfect model test, as described in Lorenz et al. (2018). Here, a simplified perfect model test is performed on an 47 member ensemble, which includes only the first initial condition member from the SMILEs and each of the CMIP5 models ensembles (40 named models with an additional 4 members from GISS-E2-R and GISS-E2-H physics physics-version ensembles). This is done because having multiple IC members (or a SMILE) in the ensemble could bias the perfect model test, which is based on predicting one member using a weighted distribution of the rest. We use member 1 for each initial condition ensemble because, often, when multiple initial condition members are available, the first member is selected (e.g. Liu et al., 2012; Karlsson and Svensson, 2013; Sillmann et al., 2013). During the perfect model test, each member is assumed to be the "truth" once and a weighting is performed using the remaining members to predict the "true" SAT change. RMSE distances (based on nine predictors) are computed with respect to the truth for the remaining members and used in the performance weighting function (wiII) described in section 3.2. The performance weights are computed for $\sigma D$ values ranging between 0 and 2 (on 0.01 intervals). For each $\sigma D$, the weighted mean SAT change is computed and compared to the "true" SAT change. The optimal $\sigma D$ for each truth is chosen to be where the difference between the weighted mean SAT change and the true SAT change is minimized. In the few cases when the weighted mean exhibits asymptotic behavior with no clear minimum difference prior to $\sigma D = 2$, the $\sigma D$ value is selected at the point where the leveling off begins (as determined by the intersection between a threshold value and the weighted mean curve). For the nine predictor RMSE weightings, we set $\sigma D$ values to the mean of the 47 optimal $\sigma D$ values computed during the perfect model test. It is important to note that this choice is ultimately subjective and further parameter sensitivity testing is recommended in studies focused on model performance."

[Figure]

For each truth, do you calculate the weighted mean and weighted standard deviation (for each value of sigma), which you use to compute the 10-90% prediction interval supposing a Gaussian distribution, and then check whether the truth is within this interval?

Good question. In the more nuanced version of the perfect model test, the prediction interval does not assume a gaussian distribution. The 10 and 90th percentiles are computed as :

For x 1 …. x i and weights w 1 …. w i ,
W is the sum of all weights and s j is the sum of the first j weights.

For the probability p, if pW falls
(1) between s j and s j +1 , the quantile is estimated at x j+1
(2) on s j , the quantile is estimated at ½ (x j + x j +1 )

Further details are given here:
https://www.statsmodels.org/dev/generated/statsmodels.stats.weig
htsta
ts.DescrStatsW.quantile.html#statsmodels.stats.weightstats.Descr
Stats
W.quantile
https://support.sas.com/documentation/cdl/en/procstat/

I suppose that the basic "equal weighting" approach pass the test, right? Therefore, how do you decide that your weighting scheme is better than the standard democracy? You say that you choose the smallest value of sigma that passes the test, but why the smallest value of sigma should be used? Why is it preferable?

For values of sigma larger than the distances between models and
observations (in our case, for sigma values larger than about 1.3),
the weighting tends towards a standard democracy. We can often see in
the perfect model test that a stronger-than-equal weighting gets you
closer to the "true" warming than an equal weighting does, suggesting
that down-weighting some models (by setting sigma, the gaussian
width, to a value within distribution of distances) does have some
added value.

In terms of "better than the standard democracy", we find it helpful
to think of the weighting as a way to assign meaning to the
distribution. In the multi-model, multi-initial condition ensemble
used in this study, the equal weighting approach is misleading; the
SMILEs have more weight than other models because they are numerously
represented. The "one model, one vote" democracy is simply a binary
weighting where much of the available information arbitrarily gets
assigned zero weight. The weighting we explore allows us to
explicitly state that models that are significantly biased with
respect to observations are unlikely to effectively simulate future
regional climate and that models that are effectively duplicates of
one another don't get the opportunity to "stuff the ballot box".

L657: "… and set sigma s at two standard deviations below the SMILE Sij mean value". I understand the logic, but why this specific choice of two standard deviation below Sij mean value?

This was an attempt to offer guidance on how to best avoid overconfident weighting in the case of a predictor set with too much internal variability to distinguish the IC members from the models. From the sensitivity analysis, it was evident that the sweet spot for $\sigma S$ was somewhere between 0.2 and 0.3 which placed it in the lower tail of the SMILE intermember distance distributions. However, we've emphasized that rather than deal with the scenario where the distribution is sensitive to the shape parameter, it's preferable to instead choose predictors that can distinguish IC members from models:

L679-684: "Another more robust option, as discussed in the main text, is to select a set of independence predictors that explicitly differentiate inter-IC member distances from inter-model distances. In this case, $\sigma S$ should not be set to two standard deviations below the SMILE Sij mean, rather it should be set to a value greater than all IC member Sij but less than inter-model Sij (particularly differently named models). For the large-scale CLIM predictor set explored in Figure 4, $\sigma S$ can be computed based on initial condition member intermember distances as described in Brunner et al. 2019; $\sigma S$ in this instance is 0.22."